# Measurement Report: New particle formation characteristics at an urban and a mountain station in Northern China

Ying Zhou[1], Simo Hakala[2], Chao Yan[1,2,*], Yang Gao[3], Xiaohong Yao[3], Biwu Chu[4], Tommy Chan[2], Juha Kangasluoma[1,2], Shahzad Gani[2], Jenni Kontkanen[2], Pauli Paasonen[2], Yongchun Liu[1], Tuukka Petäjä[2,5], Markku Kulmala[1,2], Lubna Dada[2,6,7*]

[1] Aerosol and Haze Laboratory, Beijing Advanced Innovation Center for Soft Matter Science and Engineering, Beijing University of Chemical Technology, Beijing, China

[2] Institute for Atmospheric and Earth System Research / Physics, Faculty of Science, University of Helsinki, Finland

[3] Key Laboratory of Marine Environment and Ecology, Ministry of Education, Ocean University of China, Qingdao 266100, China

[4] State Key Joint Laboratory of Environment Simulation and Pollution Control, Research Center for Eco-Environmental Sciences, Chinese Academy of Sciences, Beijing 100085, China

[5] Joint International Research Laboratory of Atmospheric and Earth System Sciences (JirLATEST), Nanjing University, Nanjing, China

[6] Extreme Environments Research Laboratory, Ecole Polytechnique Fédérale de Lausanne (EPFL) Valais Wallis, Sion, 1951, Switzerland

[7] Laboratory of Atmospheric Chemistry, Paul Scherrer Institute, 5232 Villigen, Switzerland

*Correspondence to:* Lubna Dada: lubna.dada@helsinki.fi & Chao Yan: chao.yan@helsinki.fi

**Abstract**

Atmospheric new particle formation (NPF) events have attracted increasing attention for their contribution to the global aerosol number budget, and therefore their effects on climate, air quality, and human health. NPF events are regarded as a regional phenomenon, occurring over a large area. Most observations on NPF events in Beijing and its vicinity were conducted in populated areas, whereas observations on NPF events in mountain with few anthropogenic

emissions are still rare in Beijing (Wang et al. 2013). The spatial variation of NPF event
intensity has not been investigated in detail by incorporating both urban area and mountain
measurements in Beijing. Here, we provided NPF events characteristics in summer 2018 and
2019 at urban Beijing and a comparison of NPF event characteristics — NPF event frequency,
particle formation rate, and growth rate — by comparing an urban Beijing site and a
background mountain site separated by ~80 km from June 14 to July 14, 2019 as well as give
insights into the connection between both locations. There were no significant difference of
particle formation rates and growth rates observed during the short-term observation in 2019
and longer-term observation in summer 2018 and 2019 at the urban site. During parallel
measurements at urban Beijing and mountain background areas, although the median
condensation sink during the first two hours of the common NPF events was around 0.01 $s^{-1}$
at both sites, there were notable differences in particle formation rates between the two
locations (median of 5.42 $cm^{-3}s^{-1}$ at the urban site and 1.13 $cm^{-3}s^{-1}$ at the mountain site during
the first two hours of common NPF events). In addition, the particle growth rates in the 7-15
nm range for common NPF events at urban site (median of 7.6 $nm.h^{-1}$) at the urban site were
slightly higher than those at mountain site (median of 6.5 $nm.h^{-1}$). To understand whether the
observed events were connected, we compared air mass trajectories as well as meteorological
conditions at both stations. Favorable conditions for the occurrence of regional NPF events
were largely affected by air mass transport. Overall, our results demonstrate a clear
inhomogeneity of regional NPF within a distance of ~100 km possibly due to the discretely
distributed emission sources.
Keywords: atmospheric aerosols, growth rates, regional new particle formation, sulfuric acid

## 1    Introduction

Atmospheric new particle formation (NPF) events resulting from the formation of clusters
and stable aerosol particles from gas-phase precursors have been recognized as a major
contributor to the global aerosol budget (Kulmala et al., 2004; Zhang et al., 2012). Once the
newly formed particles grow to certain sizes, they can act as cloud condensation nuclei
(CCN), affecting the regional and global climate (Pierce and Adams., 2009;Yu and Luo.,
2009). NPF events were also found to contribute to haze formation and thus can influence air
quality, especially in megacities where the precursor concentrations and associated particle
formation rates are rather high (Guo et al., 2014; 2020;Kulmala et al., 2021;Du &Dada et al.,

58     2021).

The occurrence of NPF events is a result of the competition between factors promoting and
inhibiting cluster formation and their growth. For instance, sufficient sulfuric acid and other
low-volatility vapors have been confirmed to be important in particle nucleation and growth
in field observations as well as in chamber experiments (Ehn et al., 2014;Wang et al.,
2017;Lehtipalo et al., 2018;Yao et al., 2018;Deng et al., 2020b). On the other hand,
background particles can inhibit new particle formation by acting as condensation sink for
vapor precursors and coagulation sink for newly formed particles. Indeed, Cai et al. (2017)
found that the Fuchs Surface Area ($A_{Fuchs}$) (which is linearly proportional to condensation
sink) determined the occurrence of NPF events in urban Beijing. In the atmosphere, ambient
conditions, such as air mass trajectories and meteorological conditions, can affect the
occurrence of NPF events by modifying the source-sink competition. Wu et al. (2007)
summarized favorable conditions for NPF events in Beijing based on a one-year observation
as sufficient solar radiation (sunny days), northerly wind, low relative humidity, and less
pre-loading large particles. Similarly, in other environments, plenty of radiation, intermediate
temperatures and low condensation sink favor the occurrence of NPF events (Qi et al.,
2015;Dada et al., 2017;Kerminen et al., 2018). Regional NPF events can happen with a
spatial extent up to several hundred kilometers and vertical extent from boundary layer to free
troposphere under favorable conditions (Hussein et al., 2009;Shen et al., 2011;Dai et al.,
2017). Earlier studies have shown that regional NPF events by simultaneous observations at
two or more sites had similar features in their occurrence and characteristics. For instance,
Komppula et al. (2006) investigated the occurrence of NPF events at two forest stations in
northern Finland during 2000-2003. Their results suggested that same air mass source regions,
favorable weather conditions and clean air at both stations were necessary for NPF events
occurring simultaneously at the two stations. Vana et al. (2016) compared observations at
three sites over 1000 km distance at northern Finland, southern Finland and Estonia in
2013-2014. They found that some events have the same origin. On the other hand, Jun et al.
(2014) observed that NPF events occurred less frequently at downtown Toronto than at a
nearby background site, and attributed this observation to the high condensation and
coagulation sink due to primary particle emission from traffic at urban areas. Moreover,
Carnerero et al. (2018) observed horizontal distribution and regional impact of the NPF
events with data from three urban, urban background, and suburban stations in the Madrid
metropolitan area, Spain in July 2016. Their results indicated that ultra-fine particles were
detected quasi-homogenously in an area spanning at least 17 km horizontally and the NPF
events extended over the full vertical extension of the mixed layer. Finally, Salma et al. (2016)
found that regional NPF events were modified and transformed by urban NPF events during
their observation in 2008-2009 and 2012-2013 in Budapest and at a regional background site
71 km away from it.
In comparison to the aforementioned studies in Europe, a similar study was also carried out to
understand the regional NPF events in North China Plain. Wang et al. (2013) characterized
the NPF events observed at an urban Beijing site and a regional background site about 120
km northeast to the urban site from March to November in 2008. They observed 96 and 87
NPF events at urban Beijing and background site, respectively, among which 52 NPF events
were observed simultaneously at both sites. They found that NPF events were slightly weaker
in the background site compared to those observed at the urban site. However, the factors that
influence the occurrence of NPF events at the two stations simultaneously were left
undetermined. In addition to largely populated urban areas, there is a large mountain area
within the Beijing-Tianjin-Hebei (BTH) region, where to our best knowledge, the
characteristics of NPF events are understudied. In this study, we conducted simultaneous
measurements of NPF event characteristics at an urban site in Beijing and a background
mountain site about 80 km west to urban Beijing from June 14 to July 14 2019.
Based on our observations, we aim to (i) compare the characteristics of the NPF events
between the two sites, including the frequency, particle formation rate, and particle growth
rate; (ii) figure out the connections and differences between NPF events at these two sites; (iii)
identify the favoring conditions for regional NPF events. Due to the profound participation of
NPF events in the global aerosol number loading and air quality degradation, identifying the
conditions that promote or inhibit the occurrence of regional scale NPF events could help
minimize its adverse effects.

## 2    Experiment and methodology


### 2.1 Sites' description


Urban site: The Beijing University of Chemical Technology - BUCT (39.94° N, 116.31° E)
station is located on the fifth floor of a university building inside the west campus of BUCT.
The station is surrounded by several main roads with heavy traffic and residential areas and
thus, can be considered a typical urban station. More details of this station can be found in
Zhou et al. (2020). Observations at the urban site are continuous since January 17, 2018 and
were only interrupted for necessary instrument maintenance. The location is referred to as
'**UB**' from here after and is shown on the map in Fig. 1.
Mountain site: The Beijing Forest Ecosystem Research Station (39.96° N, 115.43° E) is
located in the west of Beijing, referred to as '**MT**' from here after, which is part of the
Chinese Ecological Research Network (CERN). It is located in the mountain areas west of
Beijing, about 80 km from the urban site; see also in Fig. 1. The altitude of the station is 1170
m above sea level and it is surrounded by forests. The closest anthropogenic activities are
associated with small villages located in the valley nearby the MT station. Observations at
MT station are from June 14 to July 14, 2019. For comparison reasons, we only used the data
collected simultaneously at both stations.

### 2.2 Instrumentation


Particle number size distribution data in the size range of 6-840 nm were collected using a
differential mobility particle sizer (DMPS) at the UB station. The instrument consists of one
Hauke-type DMA (differential mobility analyzer, home-built by university of Helsinki) in
different flow rates and one CPC (condensation particle counter, TSI Model 3772). Details of
this instrument can be found in Salma et al., 2011 and Kangasluoma et al. (2020). At MT
station, a scanning mobility particle sizer (SMPS, consists of a TSI Differential Mobility
Analyzer model 3081) and a fast mobility particle sizer (FMPS, TSI Model 3091) were used
to measure particle number size distribution from June 14 to June 28 and from June 29 to
July 14, respectively. The size ranges of the SMPS and FMPS are 7-1218 nm and 6.04-856
nm, respectively. The total number concentration from 4-3000 nm, measured by
Condensation Particle Counter (CPC; TSI Model 3775), was used to calibrate the particle
number size distributions from FMPS according to the method suggested by Zimmerman et
al. (2015). More details about the instrument are found in the previous studies (Wang et al.,
2019; Gao et al., 2020). The full campaign particle number size distributions at both sites are
shown in Fig. 2. The particle number size distribution measured by FMPS correlated well
with SMPS after being calibrated (Lee et al., 2013).
To ensure high quality of particle number size distribution data at UB site, a particle number
size distribution system (PSD) also sampled in parallel with DMPS from June 1 to August 31,
2019 (summer 2019). It included a nano-scanning mobility particle sizer (nano-SMPS, 3–55
nm, mobility diameter), a long SMPS (25–650 nm, mobility diameter) and an aerodynamic
particle sizer (APS, 0.55–10 µm, aerodynamic diameter). Details of this instrument can be
seen at Liu et al. (2016) and Deng et al. (2020b).
As shown in Fig.3, median particle number size distribution obtained from PSD and DMPS
matched well with each other within a factor of 2 during our observation in summer 2018 and
2019 at UB site. We cannot compare particle number size distribution data obtained from
DMPS, SMPS and FMPS as we did not sample with these three instruments in parallel at the
same site. However, it reasonable to assume that particle number size distribution obtained
from FMPS were comparable with those from DMPS as on one hand the measurement
techniques of particle number size distribution in the size range of these two instruments have
been well developed and be applied in quite a lot observations (Wang et al., 2017;
Kangasluoma et al., 2020), on the other hand, the FMPS was carefully calibrated and
properly operated during the observation as discussed above. Similar conclusions apply for
the SMPS as well where we can rely on using the measurement from this instrument to
discuss at least NPF event frequency at MT site during June 14 to June 28, 2019, during
which parameters of only one NPF event are calculated.
Sulfur dioxide (SO₂) concentration data were collected by Thermo Environmental Instrument
model 43i-TLE with a time resolution of 5-min at the UB station. There were no direct
measurement of SO₂ concentrations at the MT station, but the SO₂ measurement at the closest
national monitoring station (Longquan station, around 60 km from MT station and 20 km
from UB station, see Fig. 1) was used to indicate the strong decline of SO₂ concentration
from urban Beijing towards the west areas. Time series of SO₂ concentration at UB station
and Longquan station during the whole observation is shown in Fig. 4. Due to the lower
emission, the SO₂ concentration at the MT station is expected to be even lower than that in
Longquan station.
The sulfuric acid concentration was measured at UB station by a chemical
ionization-atmospheric interface-time of flight mass spectrometers (CI-APi-ToF, Aerodyne
Research Inc.) equipped with a nitrate chemical ionization at UB station (Lu et al.,
2019).There were no sulfuric acid data available at MT station and since no SO₂
concentrations were available, a sulfuric acid proxy concentration could not be derived.
The meteorological conditions such as relative humidity (RH, %), temperature (℃) and solar
radiation (UVA and UVB, W/m$^2$) were measured using a Vaisala Weather station data
acquisition system (AWS310, PWD22, CL51), Metcon at UB station and using Vaisala
MAWS301 automatic weather station at MT station. The measurements at the MT station
were carried out at the height of 1.5 m. The wind speed (m/s) and wind direction (°  ) data is
also measured by the weather station at UB site, while at MT site, we obtained with
reanalyzed data from ERA5 model (Olauson, 2018).
*2.3 Air mass back trajectories*
Air mass back trajectories were calculated using a Lagrangian particle dispersion model
FLEXPART (FLEXible PARTicle dispersion model) version 9.02 (Stohl et al., 2005). As the
meteorological input, we used ECMWF (European Centre for Medium-Range Weather
Forecast) operational forecast data with 0.15° horizontal and 1-hour temporal resolution.
Particle retroplume simulations were performed hourly for both sites during the whole study
period. For each retroplume simulation, we used 50 000 model particles distributed evenly
between 0–100 m above the measurement site. The released model particles were traced
backwards in time for 72 h, unless they exceeded the model grid (20–60°N, 95–135°E,
resolution: 0.05°).
Based on the arrival direction of the 72-h backward trajectories, the prevailing air mass
transport conditions at each site were classified into 5 groups: North group, West group, East
group, South group and Local group. Air masses arriving from north, north-west and
north-east including Mongolia, Inner-Mongolia and north-east China were classified into the
North group. Air masses from Shanxi province, Inner-Mongolia and further west were
classified into the West group. Air masses from the ocean east of Beijing were classified into
the East group and air masses from southern areas were classified into the South group.
Stagnant air masses that had only travelled short distances and/or were circulating around the
measurement site were classified into the Local group. Examples of air mass trajectories
belonging to these five groups are shown in Fig. 5. In general, air masses from the north and
west supply clean air from the mountainous areas to both stations, whereas air masses from
the east and south travel over highly populated areas, thus accumulating air pollutants.
However, the impact of local air masses on the pollution levels at the two sites can be
different; at UB station, local air masses are polluted by the urban emissions, while at MT
station stagnant air could cause a clean situation due to low local emissions. More details on
the relationship between air mass transport conditions and the extent of pollution is discussed
in later sections.
*2.4 Estimating the spatial extent of NPF*
The observation of regional new particle formation events, where the growth of newly
formed particles can be followed for several hours, is a result of NPF taking place over a
large spatial area. This is because as time progresses, the particles observed at a measurement
site must have originated from further and further away due to non-zero wind conditions.
Following the progression of the observed NPF event and using air mass back trajectories, we
can estimate where the particles observed at different stages of the NPF event were initially
formed by calculating the air mass locations at the onset time of the NPF event (assuming
that NPF occurs simultaneously over the larger area). Typically, the mode related to the NPF
event disappears from the observations after some time. This is an indication of the currently
observed air mass arriving from an area where NPF was no longer taking place due to
unfavorable local conditions. If the shift in the air mass origin towards unfavorable conditions
occurs gradually over time, the mode related to the NPF event can enter a stage of growth
stagnation (or even decrease in size) before disappearing completely (Kivekäs et al., 2016).
This is because the increasing transport time between NPF onset and observation of the
particles at the measurement site provides less and less additional 'material' for aerosol
growth towards the more unfavorable conditions. Calculating the locations where NPF is
assumed to have taken place for longer data sets including several regional NPF events can
give an estimation of the typical spatial extent of NPF around the measurement location. It
should be noted that even in relatively clear cases, the subjective determination of NPF event
onset and end times can easily lead to uncertainties of few tens of kilometers in the
estimations. In locations with strong primary pollution sources, such as urban Beijing,
objective determination of said times becomes even more difficult. More details and
discussion related to the method and its uncertainties can be found in Kristensson et al.

241    (2014).

### *2.4  NPF event classification*

Particle number size distribution data from both stations were used for classifying individual
days into new particle formation (NPF) event days and non-event days. This classification
followed procedures presented by Dal Maso et al. (2005) and later adapted for urban
locations (Chu et al., 2021) in which a day is classified as a NPF event day if (a) a new mode
in the size range smaller than 25 nm appeared and (b) the new mode kept growing over
several hours. On the other hand, non-event days are the days which do not fit any of the
abovementioned criteria and undefined days are the days which fit either one of the
abovementioned criteria or the days which we cannot distinguish whether the new mode was
from NPF event or traffic.

### 2.5 Characteristics of NPF events

#### 2.5.1   Condensation sink

The condensation sink (CS) was calculated from particle size distribution data using the
method described by Kulmala et al. (2012):
$$CS = 2\pi D \sum_{dp'} \beta_{m,dp'} dp' N_{dp'}$$

257   (1)

where $D$ is the diffusion coefficient of the condensing vapor, sulfuric acid in our case, and
$\beta_{m,dp'}$ represents the transition-regime correction, $N_{dp'}$ is the particle number concentration
with diameter $dp'$. As shown in Fig. 6, particles in size range of 20-800 nm dominated the
total CS at UB station and particles in the size range of 50-800 nm dominated the total CS at
MT station. Although the size ranges of DMPS, FMPS and SMPS slightly differ, all of them
cover the main size range which constituted the CS and thus the calculation of CS should not
be significantly influenced by differences in the instrument size ranges.

#### 2.5.2   Particle growth rates

Particle growth rates were calculated for the size range of 7-15 nm ($GR_{7\text{-}15\ nm}$) using the 50%
appearance time method introduced by Lehtipalo et al. (2014) and Dada et al. (2020a)
according to
$$GR = \frac{dp_2 - dp_1}{t_2 - t_1}$$

270       (2)

where $t_2$ and $t_1$ are the appearance times of particles with sizes of $dp_2$ and $dp_1$, respectively.
The appearance time is defined as the time at which the concentration of particles at size $d_p$
reaches 50% of its maximum.
*2.5.3   Particle formation rates*
The formation rates of particles of diameters 7 nm ($J_7$) were calculated from particle number
size distribution data using the method presented by Kulmala et al. (2012) and modified for
urban environments by Cai and Jiang (2017):

$$J_k = \frac{dN_{[d_k,d_u)}}{dt} + \sum_{d_g=d_k}^{d_{u-1}} \sum_{d_i=d_{min}}^{+\infty} \beta_{(i,g)} N_{[d_i,d_{i+1})} - \frac{1}{2} \sum_{d_g=d_{min}}^{d_{u-1}} \sum_{d_i^3=\max(d_{min}^3, d_k^3-d_{min}^3)}^{d_{i+1}^3+d_{g+1}^3 \le d_u^3} \beta_{(i,g)} N_{[d_i,d_{i+1})} N_{[d_g,d_{g+1})}$$

$$+ \frac{dN}{dd_i}\bigg|_{d_i=d_u} \bullet GR_u$$


279        (3)

Here, $J_k$ is the particle formation rate at size $d_k$, $cm^3 \cdot s^{-1}$, (7 nm in this study); $d_u$ is the upper
size limit of the targeted aerosol population (10 nm in this study); $d_{min}$ is the smallest particle
size detected by particle size spectrometers (to make the results comparable, the $d_{min}$ was set
to 7 nm); $N_{[dk,du)}$ is the number concentration of particles from size $d_k$ to $d_u$; $d_i$ represents the
lower limit of the $i^{th}$ size bin; $\beta_{(i,g)}$ is the coagulation coefficient for the collision of two
particles with the size of $d_i$ and $d_g$; and $GR_u$ refers to the particle growth rate at size $d_u$, $nm \cdot h^{-1}$
Deng et al. (2020b).

### 3 Results and discussion

#### 3.1 NPF event frequencies at both stations

During our observation in summer 2018 and 2019 (from June to August of each year) at UB station, there were 155 days with valid data, 53 days of which were classified as NPF event days, corresponding to an NPF event frequency of 34%. This NPF event frequency was consistent with observations in urban Beijing in 2004 and 2008 in summer while smaller than other seasons especially winter during those observations and the observation in UB station (Wu et al., 2007;Wang et al., 2013;Deng et al., *2020b).*

For comparison of NPF characters between UB and MT stations, a parallel short-term observation was conducted at MT station from June 15 to July 14, 2019. In Fig.2, we show the particle number size distribution and CS during our short-term observations at both stations. There were a total of 12 and 13 NPF events observed at the UB station and the MT station, corresponding to an NPF event frequency of 48% (12 of 25) and 52% (13 of 25), respectively. Only days when particle number size distribution data were valid that visual inspection of the data and the number concentrations as well as instrument status do not indicate problems in the measurements for both stations were taken into consideration in our analysis. In addition, 9 NPF events were observed at both stations on the same day (referred to as common NPF events). Detailed information on the classified NPF event and non-event days, including the particle formation rates, growth rates, as well as their associated air mass origins during the short-term observation are provided in Table 1.

In order to understand the conditions favoring NPF events at both stations, we analyzed various ambient parameters including air mass trajectories, meteorological variables, condensation sink as well as sulfuric acid concentration.

*3.1.1 Favorable air mass origin for NPF events at individual locations*

In Fig.7, we show frequencies of air masses arriving at UB station from different directions during our observation in summer 2018 and 2019. The most frequent air masses arriving at UB station belonged to the South group. During our observation in the two summers, out of

155 days were 52 days belonging to the South group and 39, 32, 9 and 23 days in air masses
belong to North, East, West and Local groups, respectively. NPF event frequency with respect
to air masses is also shown in Fig. 7. It is noticeable that air mass origin influenced the
occurrence of NPF events at UB site as the majority of NPF events occurred when the air
masses were coming from the north. During our observation in summer 2018 and 2019, 34
(out of 55) NPF events occurred in air masses from the North group and 9, 2, 2 and 6 NPF
events in the South, East, West and Local groups, respectively (Fig.7a). One prominent
feature of these air masses is their difference in CS. The CS of the air masses classified as the
North group (with median values of 0.01 s$^{-1}$ at UB station) is substantially lower than that in
other air mass classes (CS = 0.03, 0.025, 0.017, 0.03 s$^{-1}$, for south, east, west and local,
respectively), which might explain the high NPF event frequency associated with this air
mass class. During the observation from June 14 to July 14 in summer 2019, the most
frequent air masses arriving at both sites belonged to the North group as shown in Table 1.
Out of 25 days, there were 8 and 9 days belonging to the North group, at UB and MT sites,
respectively. The highest frequency of NPF events also occurred when the air masses were
coming from the north. The high NPF events frequency during our observation form June 14
to July 14 could also be attributed to the frequent air masses arriving at both sites from north
to Beijing.
As shown in Table 1, NPF events occurring simultaneously at both sites only happened when
air masses arrived at both sites from the same directions, suggesting that most of the observed
NPF events took place over the whole studied area, extending for several hundreds of
kilometers (Dai et al., 2017;Du et al., 2021). The occurrences of common NPF events also
closely connected with air mass origins that 7 (out of 9) common NPF events occurred under
air masses in the North group, with the other two NPF events in the South group.
*3.1.2 The role of condensation sink in NPF event occurrence*
Figure 8a shows the difference in CS between NPF event and non-event days during our
observation in summer 2018 and 2019(two whole summers)at UB site and short-term
parallel observations at both sites. The 'NPF1' and 'non-event1' referred to NPF and
non-event days during the two whole summers, respectively, while 'NPF2' and 'non-event2'
referred to NPF and non-event days during the short-term parallel observation period from
June 14 to July 14, 2019 at both sites, respectively. The longer-term periods are used for
confirming the representativeness of the short-term overlapping period for the whole summer.
As shown in the figure, the median CS on NPF1 or NPF2 days is equivalent for UB station
($CS_{NPF1} = 0.010s^{-1}$; $CS_{NPF2} = 0.009s^{-1}$) and less than a factor of 1.2 different between
non-event1 and non-event2 in UB station ($CS_{nonevent1} = 0.023s^{-1}$; $CS_{nonevent2} = 0.020s^{-1}$), which
confirms the representativeness of our short-term measurement period to the overall urban
Beijing summer.
Our results in figure 8a show that the median CS was ~ 0.01 $s^{-1}$ during the first 2 hours of the
NPF events, at both stations. On common NPF event days, the median CS was 0.009 $s^{-1}$ at
UB station and ~0.01$s^{-1}$ at MT station, respectively. In comparison, on non-event days,
during roughly the same time period (9:00–11:00 LT), the CS was substantially higher, with
median values of 0.02 $s^{-1}$ and 0.014 $s^{-1}$, at UB and MT stations, respectively. Figure 8b
presents the median CS during the first 2 hours of NPF events on common NPF event days
measured at both stations, and shows the high correlation between the two.
Figure 8c shows the NPF event frequency as a function of CS during our observation at UB
site in summer 2018 and 2019 and how the NPF event frequency decreased with increasing
CS. When CS was smaller than 0.01 $s^{-1}$, all days were classified as NPF event days, and when
CS was larger than 0.035 $s^{-1}$, no day was classified as NPF event day. This shows the major
role of background particles in controlling the occurrence or inhibition of NPF events as
shown in several previous studies in China and internationally (Deng et al., 2020a; Cai et al.,
2017; Kulmala et al., 2017). While we cannot present a similar figure from the MT station,
the same conclusion applies where CS does play a role in inhibiting NPF observation owing
to the difference in the CS values observed between NPF and nonevents at MT as shown in
figure 8a. Yet, since the overall preexisting particle concentration at the MT is rather on the
low end, the role of CS might not be as vital at the MT station as for the UB station.
Different from NPF events under low CS (<0.01 $s^{-1}$), these NPF events under high CS were
characterized by a relatively high $H_2SO_4$ concentration ($>10^7$ cm$^{-3}$) or low particle formation
rates (Fig.9a), discussed in further details in the coming sections. In comparison, at MT
station, when CS was smaller than ~ 0.013 s$^{-1}$, most (10 out of 14) days were classified as
NPF event days as shown in Fig. 9d. When CS was larger than ~0.013 s$^{-1}$, we only observed
one local NPF event and another two non-local NPF events (Table 1). The local NPF event
under high CS at the MT station was characterized as high UV ($>30$ W/m$^2$) and low
formation rate ($J_7$ were too small to be reliably calculated) as well.
*3.1.3 Role of meteorological variables in NPF event occurrence*
While the air mass source regions, and their connection to the CS, seem to explain the general
picture of NPF event occurrences at the two sites well, we still have some cases unexplained.
For example, as shown in Table 1, there were several non-event days observed at MT station
with air masses belonging to North and West groups, which were connected to low CS. This
indicates that a further investigation into other NPF-related variables is still required.
In Figure 10, we show diurnal variation of meteorological variables during our observation in
summer 2018 and 2019 at UB site and observations from June 14 to July 14 in 2019 at UB
and MT sites. It is noticeable that the short-term observation compared well with the
long-term observation and therefore is representative of summer at UB site as shown in
Fig.10.
First, the intensity of solar radiation is considered one of the most important parameters
deciding NPF event occurrence as it translates into photochemistry strength (Chu et al., 2019).
The median UV (UVA+UVB) intensity at the UB station on NPF event and non-event days
was 38.3 and 32.9 W/m$^2$, respectively. The UV intensity was on average ~15% higher on
NPF event days than on non-event days at UB station. Although UV intensity was important
for NPF event occurrence, we still observed NPF events at UB station under low UV intensity,
e.g. cases on June 30 and July 6. These two events all started immediately after sunrise (6:30
LT on June 30 and 7:00 LT on July 6, see Table 1) and median UV intensity during the first
two hours of NPF events was only 13.2 and 14.1 W/m$^2$, respectively. However, sulfuric acid
concentration was higher than $10^7$ cm$^{-3}$ at the same time, the possible reason is high $SO_2$
concentration and low CS (~0.003 s$^{-1}$) outcompeted the low UV intensity (Dada et al., 2020b)
as well as the possibility of having other $H_2SO_4$ sources (Yao et al., 2020).
At MT station, the median UV intensity on NPF event and non-event days was 28.4 and 14.2
W/m$^2$, respectively. The lower UV at MT station, in general might be related to the higher
RH (Fig. 10c&d) and thus more cloudiness and fog at the MT station (Hamed et al.,
2010;Dada et al., 2018). The UV intensity was on average ~100% higher on NPF event days
than on non-event days at UB station. All local NPF events happened when UV intensity was
higher than 15 W/m$^2$ as shown in Fig. 9d.
On the other hand, as shown in Fig. 10c&d, the median relative humidity (RH) was lower on
NPF event days than non-event days at both stations. This is consistent with earlier results
that high RH suppressed NPF events by increasing CS and coagulation sink (CoagS), as it
can enhance the particle hygroscopic growth (Hamed et al., 2010; Hamed et al., 2011). In
addition, high RH was also found to be associated with more clouds resulting in less solar
radiation (Dada et al., 2018).
The median temperatures at UB on event and non-event days were 31 ℃ and 29 ℃,
respectively, and at MT station 23 ℃ and 19 ℃, respectively. The median temperature was
lower at the MT station than at the UB station, due to the higher altitude of the station and
likely also the weaker solar radiation (Fig. 10e&f). At both stations, the median temperature
was very similar on NPF event and non-event days, suggesting that temperature was not a
crucial factor for NPF event occurrence during the measurement in summer.
*3.1.4 Role of sulfuric acid concentrations in NPF event occurrence*
Sulfuric acid has been found to be the main precursor vapor participating in NPF in China
and in many locations around the world due to its low volatility (Yao et al., 2018;Chu et al.,
2019). In Fig. 9a, we show the concentration of sulfuric acid as a function of CS during
summer 2018 and 2019 at UB site. As shown in Fig. 9b, the median sulfuric acid ($H_2SO_4$)
concentrations at UB station were $8.1×10^6$ cm$^{-3}$ and $4.5 ×10^6$ cm$^{-3}$ on NPF event days and
non-event days, respectively, during observation from June 14 to July 14 in 2019 and $7.9×10^6$
cm$^{-3}$ and $3.4 ×10^6$ cm$^{-3}$ on NPF event days and non-event days, respectively, during the
observation in summer 2018 and 2019. This suggests that $H_2SO_4$ was important for NPF
events at the UB station (Deng et al., 2020b; Dada et al., 2020b). On the other hand, as shown
in Fig.9a, the $H_2SO_4$ concentration during 9:00- 11:00 (local time) on non-event days could
be comparable with that on NPF event days, especially when CS was high. The $H_2SO_4$
concentration during 9:00- 11:00 (local time) on non-event days could be comparable with
that on NPF event days, especially when CS was high. Altogether, our observation shows that
the occurrence of NPF events was controlled by both $H_2SO_4$ and CS at the UB station (Cai et
al., 2020).
In addition, although we did not perform the measurement of $H_2SO_4$ at the MT station,
concentration of $H_2SO_4$ is expected to be much lower than that at the UB station. First, the
$SO_2$ concentration at measurement at Longquan Town was always below the detection limit
(~ 0.5 ppb) during our observation period. In comparison, median $SO_2$ concentration at UB
station was 0.87 ppb for all days and 0.65 ppb for NPF event days during our short-term
parallel observation period. The spatial decreasing trend of $SO_2$ concentration from urban
Beijing to the west implies a low $SO_2$ concentration at the MT station, especially when the
nearby anthropogenic sources are sparse (Liu, 2008;Ying, 2010;Wang, 2011;Yang-Chun et al.,
2013). Second, the oxidation of $SO_2$ by photochemistry reactions could also be limited by the
low solar radiation at the MT station as we discussed in 3.1.3. Third, CS, as the main sink of
$H_2SO_4$, was comparable at the MT station to that in the UB station on NPF event days (as
shown in Fig.8a). Altogether, the lower production rate and the equivalent loss rate of $H_2SO_4$
at the MT station likely results in the lower $H_2SO_4$ concentration, in comparison to UB
station.
Due to the lack of $H_2SO_4$ measurement, the NPF mechanism at the MT station cannot be
inferred. Nevertheless, we show that the occurrence of NPF as a response to photochemistry
(and very likely to $H_2SO_4$) and CS in Fig. 9d. It is clear that high UV intensity and low CS
favored the occurrence of NPF. However, there existed exceptions. For example, two NPF
events were observed even when the UV intensity was low and the CS was high, besides, it
was an undefined day on June 28 despite of the high UV intensity and low CS. These
exceptional cases will be discussed in detail in Section 3.6.1 and 3.6.2, respectively.
*3.2 NPF event start time at both stations*
There was no significant difference in NPF event start time between the long-term and
short-term parallel observations at UB site. In this section, we only compare NPF event start
time of common events at UB and MT sites during the short-term parallel observations.
During our observation, there was no advection of air masses between the two sites on
common NPF event days, indicating that the NPF events occurred at each site independently.
As shown in Table 1, all common NPF events started after sunrise and prior to noon except
the two non-local NPF events at MT station. However, NPF event start time was different
between the two sites. Earlier researches in Nanjing, China and Nordic stations showed the
similar results that NPF events can be observed simultaneously at two or more sites, but the
start time can be different, local meteorology, source strength and background aerosols could
drive temporal behavior of NPF events at each sites (Hussein et al., 2009;Dai et al., 2017).
*3.3 Particle formation and growth rates at both stations*
The particle formation rates ($J_7$) at the two stations during the measurements are presented in
Fig.11a. $J_7$ observed during the short-term parallel observation (NPF2) at UB site was in the
range of 3.0-10.0 $cm^{-3}\,s^{-1}$ with a median of 5.4 $cm^{-3}\,s^{-1}$, comparable with those observed in
summer 2018 and 2019 (NPF1 = 2-14.0 $cm^{-3}\,s^{-1}$ with a median of 4.9 $cm^{-3}\,s^{-1}$) and
significantly higher than the values in the MT station (0.75-3.0 $cm^{-3}\,s^{-1}$ with a median of 0.82
$cm^{-3}s^{-1}$) for common NPF events (Fig.11b). These values are comparable to earlier
observations in urban Beijing and another regional background station in North China Plain
(NCP) (Wang et al., 2013). Earlier observations in NCP and Yangtze River Plain also
observed higher formation rates at urban sites than corresponding background sites by
roughly a factor of 2 due to lower anthropogenic emissions at background sites (Wang et al.,
2013;Dai et al., 2017;Shen et al., 2018). The much lower $J_7$ observed at MT station is very
likely associated with the low $H_2SO_4$ concentration at this station, as we discussed above.
However, other reasons, such as the low concentration of $H_2SO_4$ stabilizers, e.g., amines,
cannot be ruled out either. Also, the $J_7$ at UB station could be affected by particle emissions
due to the proximity of the location to the highway (Kontkanen et al., 2020).
The particle growth rates in size range of 7-15 nm ($GR_{7-15nm}$) at the UB station (4.8-12.9
nm/h with a median of 7.8 nm/h) during NPF2 was also comparable with the whole summers
(NPF1) (4.8-12.9 nm/h with a median of 8.5 nm/h). While the difference in $J_7$ was 7 times
higher in UB than in MT, the observed GR were only a slightly higher at UB than at the MT
station (5.7-10.5 nm/h with a median of 6.5 nm/h) for common NPF events (Fig.11c&d),
implying that precursors needed for particle formation were much more abundant in the
polluted urban environment (Wang et al., 2013), while those needed for growth are rather
comparable. The GR at UB station was comparable with other long-term observation at UB
station (1.1-8.0 nm/h) in 2018, and other urban areas in China (Herrmann et al., 2014;Chu et
al., 2019;Deng et al., 2020b). Consistent with earlier observations showing that $H_2SO_4$ could
only contribute to a small fraction of the particle growth at this size range (Paasonen et al.,
2018;Qi et al., 2018;Guo et al., 2020), the growth rates at both stations cannot be explained
by the $H_2SO_4$ concentration. This implies that other condensable species, very likely
low-volatility organic vapors, play an important role in particle growth at both stations. At the
UB station, anthropogenic VOCs are dominant precursors of these low-volatility organic
vapors (Guo et al., 2020;Deng et al., 2020a), while VOCs at MT station, with rare
anthropogenic sources, are likely dominated by biogenic emissions.

*3.4 Ending diameters of newly-formed grown particles*

Earlier observations have shown that diameters of newly-formed particles should be larger
than 70 nm to contribute to cloud condensation nuclei significantly (Man et al., 2015; Ma et
al., 2021) and will be considered as haze particles when their size reaches larger than 100 nm
(Kulmala et al., 2021). In Fig. 12, we show ending diameters (End Dp) of newly formed
grown particles during our observations at both sites. End Dp during the observation from
June 14 to July 14 at UB site (21-105 nm, with a median of 49 nm, Fig.12a) had similar

characteristics as those during the long-term observation in summer (21-126 nm, with a median of 56 nm, Fig.12a) where most of End Dp were in the range of 25-70 nm. As shown in Fig.12b, 61% of End Dp were in the range of 25-70 nm, and only 9% of End Dp were larger than 100 nm during our observation in summer 2018 and 2019 at UB site. We found that the ending diameters slightly higher at UB site than MT site, but the difference is not significant (49 nm vs 45 nm) as shown in Fig. R12c.

Earlier research has pointed out that in order to observe particle growth until 100 nm at a measurement station under typical conditions, simultaneous NPF should happen over a very large area (e.g. with wind speed 5 m/s and growth rate of 3 nm/h from the station to roughly 600 km upwind from the station) (Paasonen et al., 2018). During our observation in summer 2018 and 2019, most of the newly formed modes kept growing for about 20 hours after an NPF event started, and the maximum horizontal extension of the observed NPF events in the growth stage is restricted to within about 200 km (~2° in latitude) north of UB site (Fig. 13). As shown in Fig.13, the population density is also higher within the area extending ~200 km north than beyond this limit. Therefore, it seems that NPF events were limited to the regions with some contribution from anthropogenic emissions during air mass transport from north to Beijing. Roughly similar extent of the NPF area is also seen in other directions. However, towards the south it is more likely that increasing condensation sink from accumulating pollution becomes the limiting factor for NPF occurrence rather than decreasing strength in emission sources. NPF events at MT station had similar characteristics as those at UB station with the NPF event region extending a few hundred kilometers towards the north. The NPF events in this direction were disrupted after a relatively similar distance (or they enter the growth stagnation phase, which will be discussed in section 3.6.3). The limited NPF event area could possibly explain why most End Dp we observed were smaller than 70 nm.

### 3.5 Effect of topography

In Figure 14 we show average particle number size distribution and particle number concentration on NPF event and non-event days during our short-term parallel observation at

both sites. On NPF event days, nucleation and Aitken mode particle number concentrations
were much smaller at MT station than those at UB station due to smaller particle formation
rates and less anthropogenic emissions. Interestingly, accumulation mode particle number
concentrations were higher at MT station (701-2900 $cm^{-3}$, with a median of 1500 $cm^{-3}$) than
that at UB station (350-1416 $cm^{-3}$ with a median of 700 $cm^{-3}$) (Fig.14b). Due to the close
proximity of the two measurement sites, the air mass arrival directions and source regions
were (mostly) similar at both sites throughout the measurement period, hence the regional
and transported cannot explain the higher accumulation mode particle number concentration
at MT site. As there were few primary emissions at MT site, the accumulation mode particles
could be attributed to secondary particles (Kulmala et al., 2021), indicating particles at MT
station were more aged than those at UB station (Fig.14a). The possible reasons is that
mountains block pollution diffusion, which in the end resulted in comparable CS at MT
station as UB station.
Figure 15 shows an example of the wind distribution before and during NPF event on June
30, 2019 at 850 hPa (close to the altitude of MT station) and 10 m above ground level. As
shown in Fig.15, the reanalyzed wind directions at 850 hPa were similar as those at 10 m
above the ground level at MT station. Actually, the wind conditions on other NPF event days
at MT station during our observation had similar characteristics that the wind directions were
similar between 850 hPa and 10 m above ground level indicating air masses well mixed
during NPF events. Earlier observations also found NPF event happened uniformly within the
mixing layer at their observation stations and particle number size distribution remains
roughly constant within the mixing layer (Shen et al., 2018;Lampilahti et al., 2021).
*3.6 case studies*
*3.6.1 Non-local NPF events at MT station*
As we discussed above, NPF events at MT stations were favored by strong photochemistry
(sufficient solar radiation) and low CS. However, we also observed two NPF events under
low solar radiation and high CS on June 15 and 25. These two NPF events had similar
characteristics, and we explain the case on June 15 in detail. During this case, air masses
arrived at both stations from south-east around 9:00 LT as shown in Fig. 16b&d, resulting in
high CS especially at MT and UB stations (Fig. 16a&c). The NPF event at the UB station was
observed around 11:00 LT, with a high median $J_7$ of 5.56 cm$^{-3}$s$^{-1}$, whereas no indication of
NPF event at MT was observed until 15:00 likely due to the high CS. After 15:00 LT, a new
growing mode from 15 nm appeared at MT station. Because there was no intense increase of
sub-15 nm particle number concentration throughout the whole event, the NPF event at MT
station was not local but occurred somewhere else and transported to MT station. This a
common phenomenon, particularly when the conditions do not favor NPF events to occur on
site, but are NPF-favorable somewhere else. The particles formed off-site are transported
vertically or horizontally and observed on site (Dada et al., 2018;Leino et al., 2019). Different
from other NPF events, this non-local NPF event was associated with strong southerly wind
(Fig. 16e), the NPF event observed at the MT station might originate from urban areas 60 km
south to the station as shown in Fig.1, assuming the NPF event started around 9:00 and the
mean wind speed was 3 m/s.
*3.6.2 Undefined day under low CS and high UV at MT station*
Interestingly, we also observed an undefined day at MT station with low CS (0.006 s$^{-1}$) and
high UV (28 W/m$^2$) on June 28 (Fig. 17c) as there seems to be a very weak 'banana' around
13:00 in the particle number size distribution plot. All other days with such conditions were
classified as NPF event days. In this case, the reasonable explanation would be low precursor
vapours which we think are $SO_2$ in our case. On this day, an air mass from Inner-Mongolia
arrived at both stations, resulting in very low $SO_2$ concentration at the UB station among all
NPF event days during our observation as shown in Fig. 4. It is reasonable to assume that the
$SO_2$ concentration was even lower at the MT station than at the UB station, and low $H_2SO_4$
concentration could also be expected, which could be insufficient to trigger an NPF event.
This is consistent with an earlier long-term observation at Shangdianzi, another background
site of Beijing, where the NPF events were suppressed by air masses from Inner-Mongolia
due to the low precursor concentrations (Shen et al., 2018). In comparison, we observed a
very weak NPF event at UB station at the same day, as local emissions were enough to supply
enough vapors to initiate NPF event.
*3.6.3 Growth stagnant and shrinkage case during our observation*
During our observations in summer 2018 and 2019 at urban site and the observation from
June 14 to July 14, 2019 at MT site, there were some cases where the newly-formed particles
entered a phase of growth stagnation or even displayed a decreasing mode diameter. On June
30, 2019 such case occurred simultaneously at both sites and we chose this day for a case
study.
As shown in Fig.18 a&b, the newly-formed particles entered a phase of growth stagnation
almost at the same time around 12:00 at both sites. Particle mode diameters decreased from
31 nm to 15 nm at UB station around 15:00, under relatively calm meteorological condition
indicating meteorological condition could not be the reason for particle sizes decrease on
June 30, 2019 at UB site. Around the same time, mode diameter at MT station also decreased
gradually from 25 nm to 16 nm. Earlier observations in summer-time Beijing have speculated
mode diameter decrease to be related to particle evaporation, which is triggered by favorable
meteorological conditions and vapor dilution (Zhang et al., 2016). From Fig.13, we see, that
the air masses observed during the growth stagnation or diameter decrease (both marked
under growth stagnation in the figure) were often located quite far in the north over the less
populated areas during the onset time of regional NPF. It is also possible, that the less
favorable initial conditions for particle formation and growth over these areas, combined with
increasing wind speed or temporal changes in the growth rate, could explain the observations
of decreasing particle sizes without evaporation (Kivekäs et al., 2016; Hakala et al., 2019).
**4   Summary and conclusions**
We conducted observations of NPF events at an urban site (UB) and a background mountain
site (MT) in Beijing and fully analyzed the favorable conditions for NPF event occurrences at
each of the sites. In order to identify the similarities and differences between NPF events at
both stations in terms of frequency, intensity and mechanisms, we compared certain NPF
characteristics including particle formation rate, growth rate as well as NPF event start time
and ending diameters of newly-formed growing particles at both stations. We found that NPF
events are most of the time a regional phenomenon occurring over the studied areas and
connected closely with air masses source regions during our observation. The air masses from
north favored common NPF events more than any other mass trajectories due to their
associated clean air masses and thus low CS. Additionally, air masses from the north group
always resulted in an NPF event at UB station, while other factors still suppressed their
occurrence at the MT station. For example, we found that sufficiently high solar radiation, e.g.
UV (UVA+UVB) intensity larger than 15 $W/m^2$ is required for an NPF event to occur at MT
and NPF events observed under solar radiation conditions smaller than 15 $W/m^2$ were rather
transported NPF events from areas upwind to MT station. Another factor suppressing the
occurrence of NPF events at MT is the too low precursor gas concentrations (e.g. $SO_2$) which
was visible in MT rather than at UB.   Moreover, we found that the CS limit for NPF event
occurrence at UB station was ~0.032 $s^{-1}$, which is consistent with earlier observations in
urban Beijing. In comparison, at MT station the CS limit could be only ~0.013 $s^{-1}$, above
which local-NPF events could possibly be suppressed associated with the lower $SO_2$
concentration.
Although NPF events could happen simultaneously at both stations, the NPF event strength
(particle formation rates) was significantly higher at UB than MT station, possibly due to
spatial inhomogeneity in the sources of aerosol precursor compounds as well as solar
radiation. In addition, the particle growth rates in size range of 7-15 nm were also slightly
higher at UB than MT station. Regional NPF events were observed to occur with the
horizontal extent within around 200 kilometers when air masses arriving at Beijing from the
north, as a result, only a few NPF events were observed to end with mode diameters larger
than 70 nm. The size of the area with NPF events upwind our observation seems to be
connected with population density that weaker NPF event was assumed to happen in less
populated areas.
Overall, our results highlight the importance of anthropogenic emissions for NPF events
occurrence and subsequently growth in north China plain during summer. However, there are

still some uncertainties due to the limited data set. For more robust knowledge on NPF events in north China plain, long-term and more comprehensive observations on NPF events upwind and downwind urban Beijing are important. Such observations can shed light into the regionality of NPF events and the dynamical development of the aerosol population influenced by radical chemistry in the plume of a megacity.

**Conflict of interest:** The authors declare no competing interests.

**Author contributions:** YZ, CY, YG, XY performed the measurements. YZ, SH, CY, YG, LD, XY analyzed the data. YZ, CY, SH, LD wrote the manuscript. All authors reviewed the paper and contributed to the scientific discussions.

**Data availability:** The data displayed in this manuscript will be available online at zenodo.com once the manuscript is in its final publication format.

**Financial support:** This publication has been produced within the framework of the EMME-CARE project, which has received funding from the European Union's Horizon 2020 Research and Innovation Programme (under grant agreement no. 856612) and the Government of Cyprus. This research has also received funding from the European Commission grant agreement no. 742206 ("ERC-ATM-GTP") as well as Academy of Finland Projects 316114 & 311932. Simo Hakala acknowledges the doctoral programme in Atmospheric Sciences (ATM-DP, University of Helsinki) for financial support. The sole responsibility of this publication lies with the author. The European Union is not responsible for any use that may be made of the information contained therein.

666

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

**Table 1:** NPF event and non- event days during our observation at both stations.

| Date | Type | Air masses (9:00-15:00) | | GR$_{7\text{-}15nm}$ (nm/h) | | $J_7$ (cm$^{-3}$s$^{-1}$) | | Event Start (LT) | | Ending diameter (nm) | |
|---|---|---|---|---|---|---|---|---|---|---|---|
| | | UB | MT | UB | MT | UB | MT | UB | MT | UB | MT |
| 2019/06/14 | a | North | North | 8.61 | - | 4.97 | - | 9:00 | 8:00 | 71 | - |
| 2019/06/15[*] | a | Local | Local | 12.63 | - | 5.56 | - | 11:00 | 15:00 | 82 | 60 |
| 2019/06/17 | d | East | Local | | | | | | | | |
| 2019/06/18 | c | Local | West | | 10.5 | | 0.17 | | 12:00 | | 45 |
| 2019/06/19 | d | South | Local | | | | | | | | |
| 2019/06/21 | d | East | Local | | | | | | | | |
| 2019/06/23 | e | East | East | | | | | | | | |
| 2019/06/24 | f | Local | Local | | 8.21 | | - | | 12:00 | | 50 |
| 2019/06/25[*] | a | Local | Local | - | - | - | - | 12:00 | 15:00 | - | 53 |
| 2019/06/28 | g | West | West | - | | - | | 11:00 | | | |
| 2019/06/29 | a | North | North | 12.93 | 7.14 | 6.93 | 2.28 | 9:00 | 8:00 | 21 | 19 |
| 2019/06/30 | a | North | North | 4.82 | 6.57 | 9.86 | 1.37 | 6:30 | 9:30 | 31 | 25 |
| 2019/07/01 | a | North | North | 7.31 | 5.82 | 3.84 | 0.82 | 9:00 | 8:30 | 105 | 102 |
| 2019/07/02 | d | Local | West | | | | | | | | |
| 2019/07/03 | a | North | North | 7.89 | 6.52 | 3.25 | 0.75 | 9:00 | 8:00 | 72 | 46 |
| 2019/07/04 | b | Local | Local | - | | - | | 10:00 | | 53 | |
| 2019/07/06 | a | North | North | 7.39 | 6.51 | 9.21 | 1.75 | 7:00 | 9:30 | 25 | 19 |
| 2019/07/07 | b | North | North | 7.61 | | 3.61 | | 9:00 | | 32 | |
| 2019/07/08 | d | East | East | | | | | | | | |
| 2019/07/09 | d | East | East | | | | | | | | |
| 2019/07/10 | h | East | East | | | | | | | | |
| 2019/07/11 | d | East | East | | | | | | | | |

| 2019/07/12 | f | East | East |  | 5.57 |  | 0.37 |  | 9:30 |  | 24 |
| 2019/07/13 | c | Local | North |  | 6.32 |  | 0.70 |  | 10:00 |  | 30 |
| 2019/07/14 | a | North | North | 12.04 | 9.86 | 3.91 | 0.89 | 9:30 | 9:30 | 63 | 47 |

'a' means NPF event observed at both stations, 'b' means NPF event day at UB station while non-event day at MT station, 'c' means NPF event day at MT station while non-event day at UB station, 'd' means non-event day at both stations on the same day, 'e' means undefined day at both stations, 'f' means undefined day at UB station while NPF event day at MT station, g means undefined day at MT station while NPF event day at UB station, h means undefined day at UB station while non-event day at MT station, * means NPF event observed at MT station was transported from somewhere else. – means the values cannot be reliably calculated. Only days when particle number size distribution were valid are included in this table.

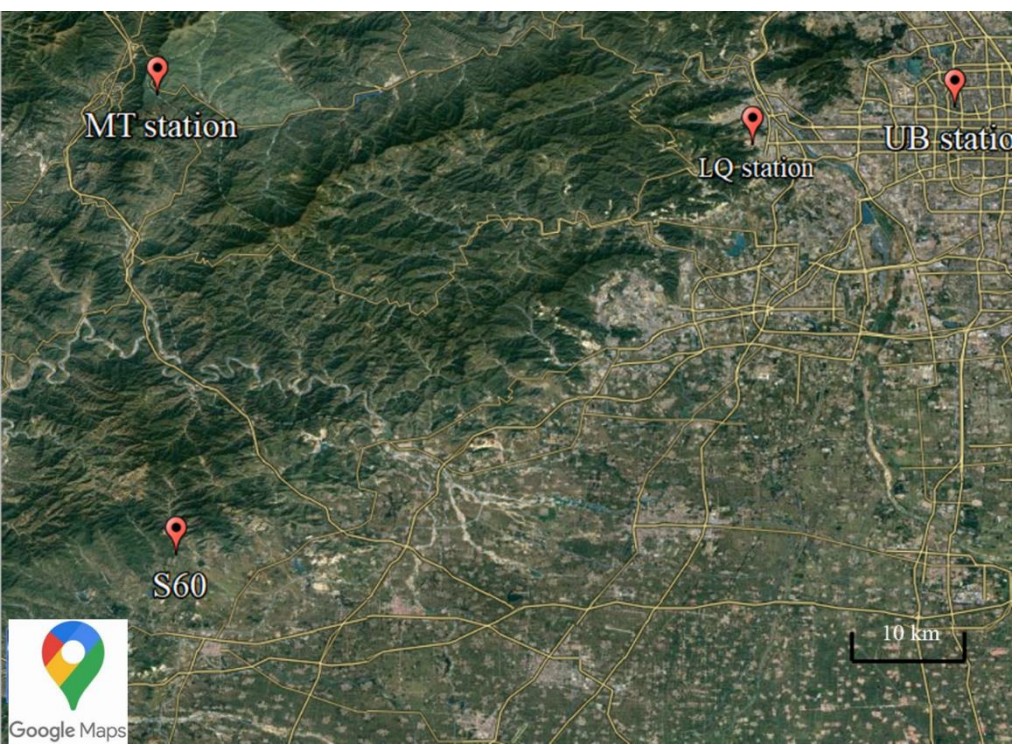


**Figure 1:** Map showing locations of urban station (UB), Longquan station (LQ), mountain
station (MT) and another site 60 km south from MT station (S60). The S60 referred to the
location where particles formed during the non-local NPF event observed at MT station on
June 15, 2019. Image is produced using © Google Maps.


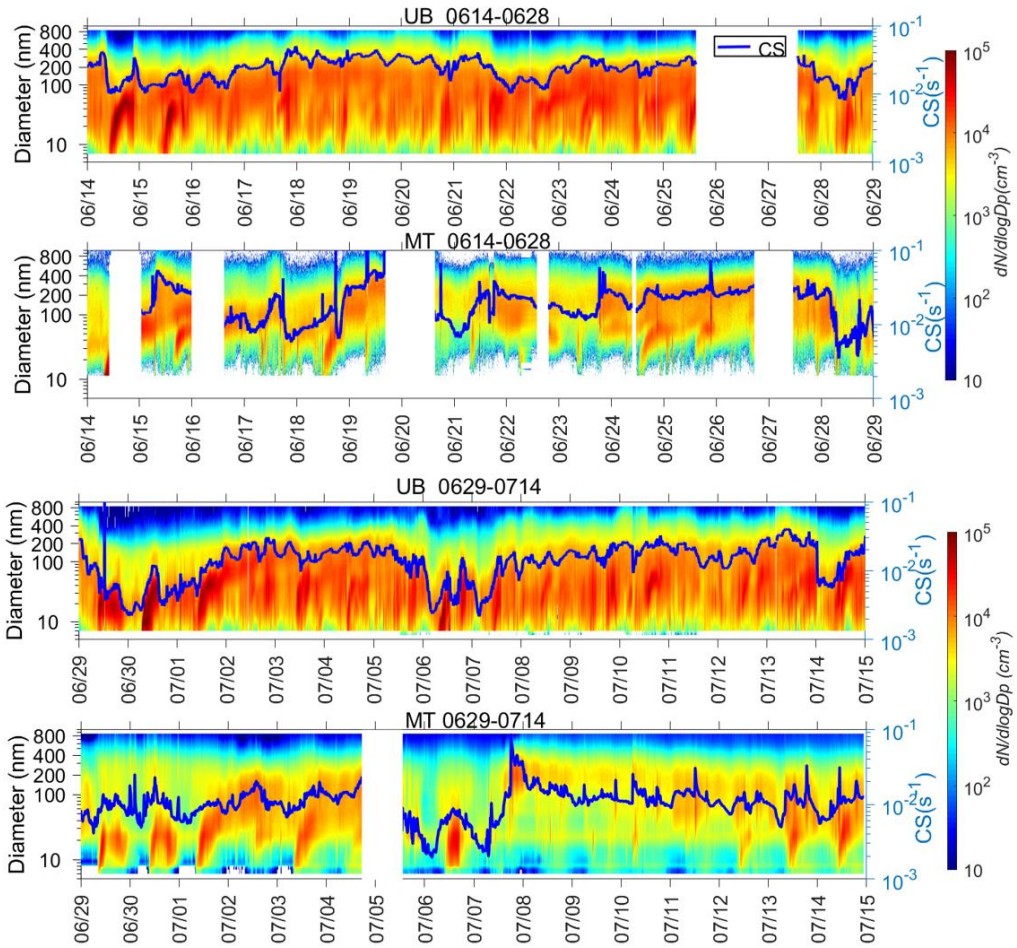

**Figure 2:** Time series of particle number size distribution and CS (the blue line) at UB and MT stations during our observations. Time resolutions for particle number size distribution data and CS were 8 min at UB station and 4 min at MT station, respectively.

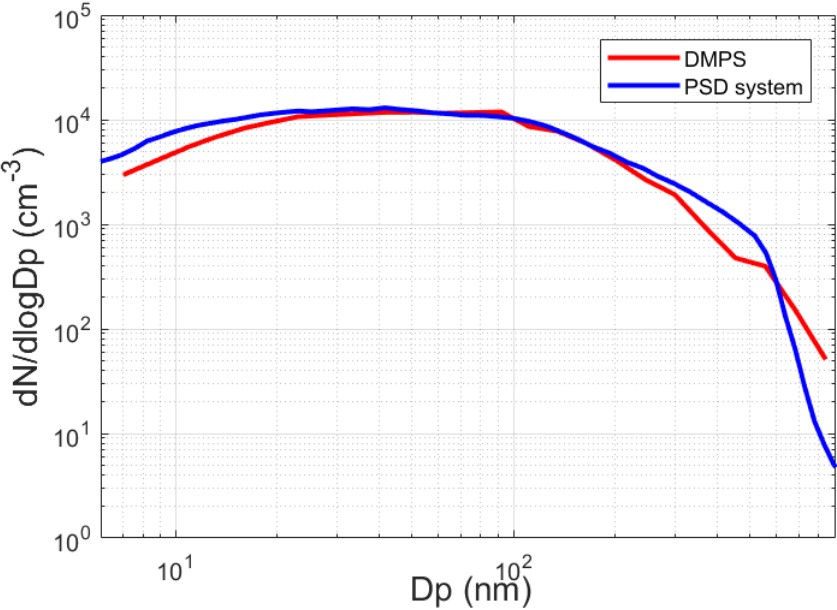


**Figure 3**: Median particle number size distribution in 5-900 nm measured by DMPS and PSD

during our observation from June 1 to August 31, 2019 at UB station. The time window of
the data is from 9:00-15:00 of every day.

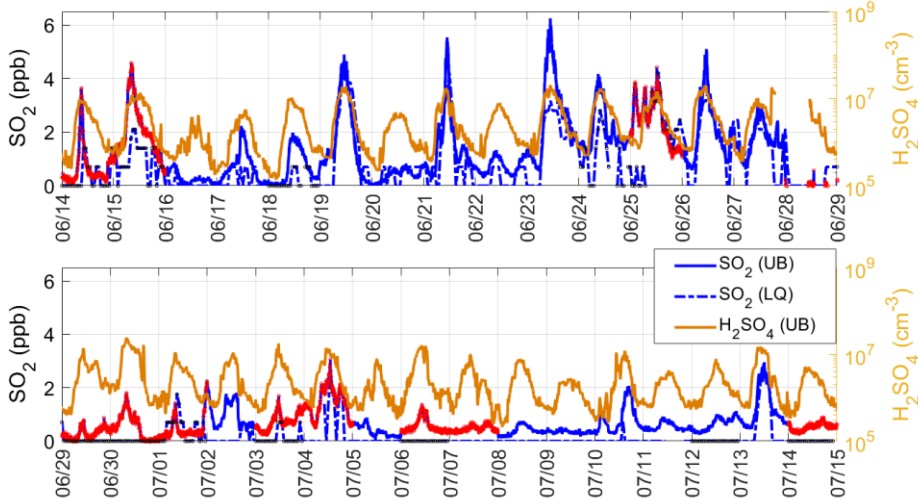


**Figure 4:** Time series of $SO_2$ concentration (ppb) at UB station and Longquan station (LQ)

during our observation (left axis) as well as $H_2SO_4$ concentration measured at UB station
(right axis). Data under detection limit are set as zero at both stations. Data on NPF event
days were marked in red at UB station and black at MT station. Time resolution for $SO_2$ data
was 5 min at UB station and 1h at LQ station, respectively.

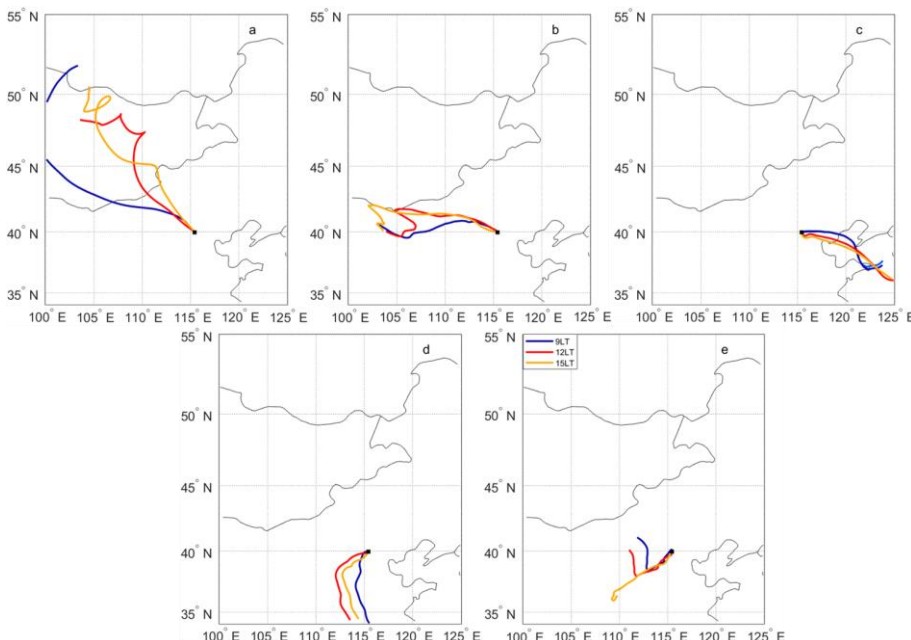


**Figure 5:** Examples of air masses arrived at both stations from (a) North group, (b) West group, (c) East group, (d) South group and (d) Local group during 9:00-15:00 (local time, LT). Both stations are under the same marker.







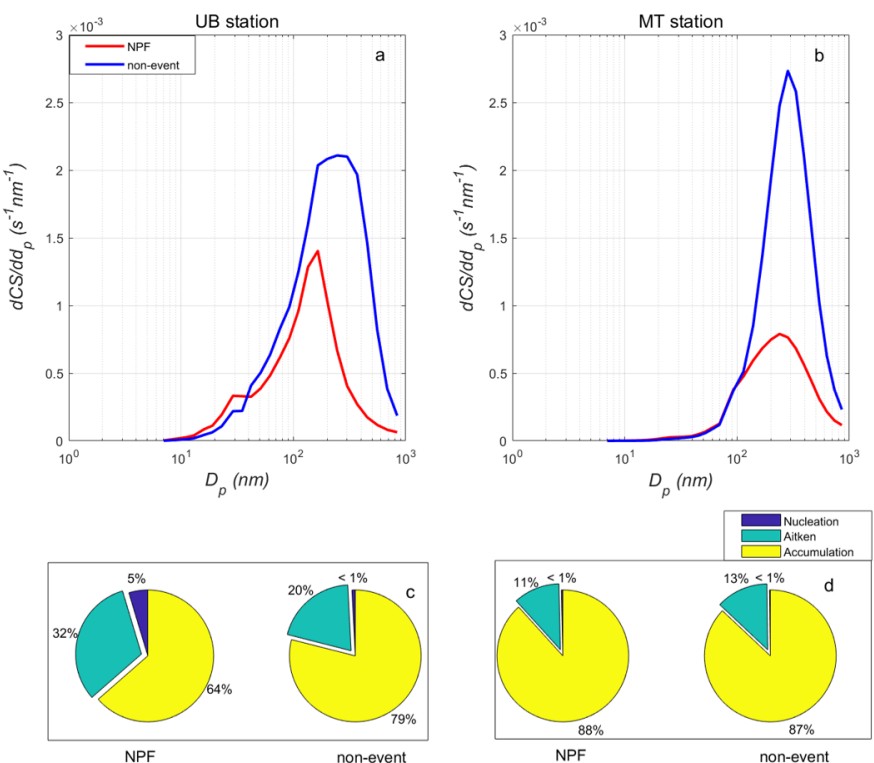


**Figure** 6**:** Median CS size distribution at UB (a) and MT (b) stations on NPF event and
non-event days, respectively during 9:00-15:00 (local time, LT) and median contribution of
nucleation, Aitken and accumulation mode particles to total CS at UB (c) and MT (d) stations
on NPF event and non-event days, respectively during 9:00-15:00 (local time, LT). The
time resolutions for CS and particle number concentration data were 8 min at UB station and
4 min at MT station, respectively.

985

986

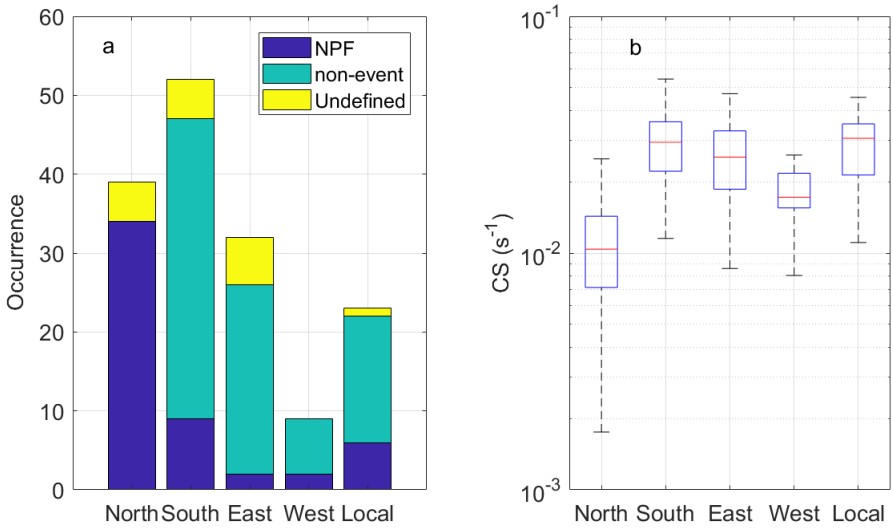

987

**Figure 7:** Occurrence of NPF events and non-events under air masses arriving from different directions (a) as well as medians and percentiles of condensation sink (CS, s$^{-1}$) during the 9:00-11:00 (local time) under different air masses (b) during our observation in summer 2018 and 2019 at UB station. The red line represents the median of the data and the lower and upper edges of the box represent 25$^{th}$ and 75$^{th}$ percentiles of the data, respectively. The length of the whiskers represents 1.5× interquartile range which includes 99.3% of the data. Data outside the whiskers are considered outliers and are marked with red crosses. The time resolution of CS was 8 min.

996

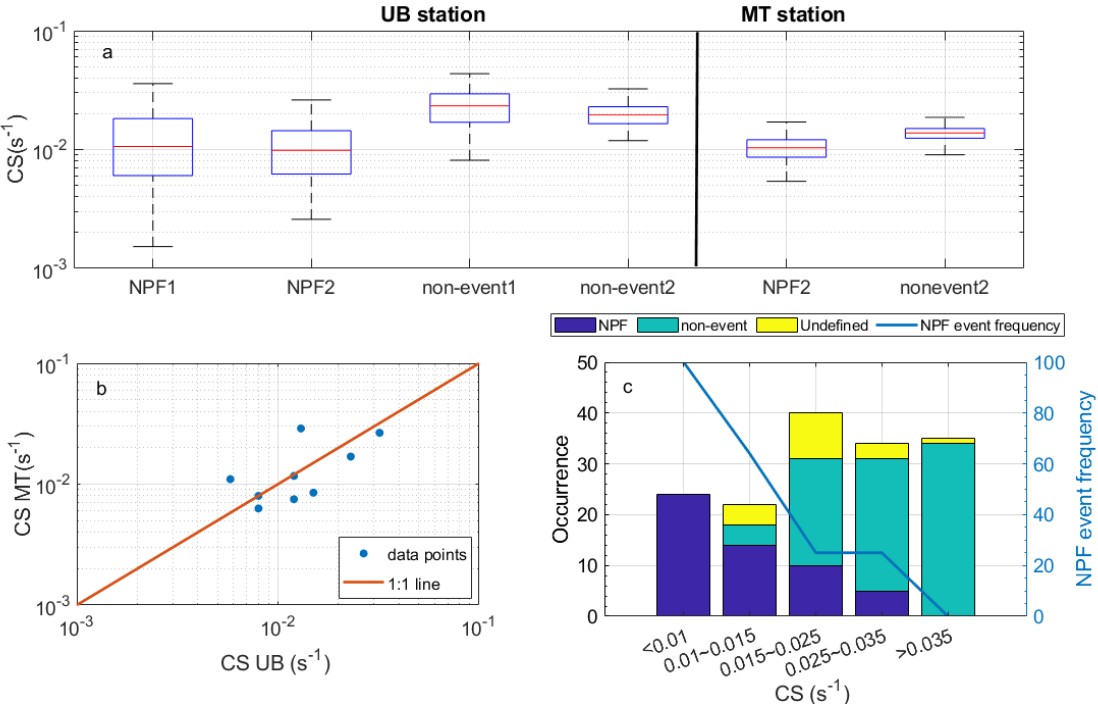

997

**Figure 8:** (a) Median and percentiles of condensation sink (CS, s$^{-1}$) during our observations at both stations. The 'NPF1' and 'non-event1' referred to NPF and non-event days during summer 2018 and 2019, while 'NPF2' and 'non-event2' referred to NPF and non-event days during the short-term parallel observation from June 14 to July 14, 2019 at both sites. The red line represents the median of the data and the lower and upper edges of the box represent 25$^{th}$ and 75$^{th}$ percentiles of the data, respectively. The length of the whiskers represents 1.5× interquartile range which includes 99.3% of the data. The time resolution of CS was 8 min. (b) Median CS during the first 2 hours of NPF events on common NPF event days measured at both stations (MT vs. UB) . (c) Numbers of NPF event, non-event and undefined days as well as NPF event frequency as a function of CS during our observation in summer 2018 and 2019 at UB station.

1009

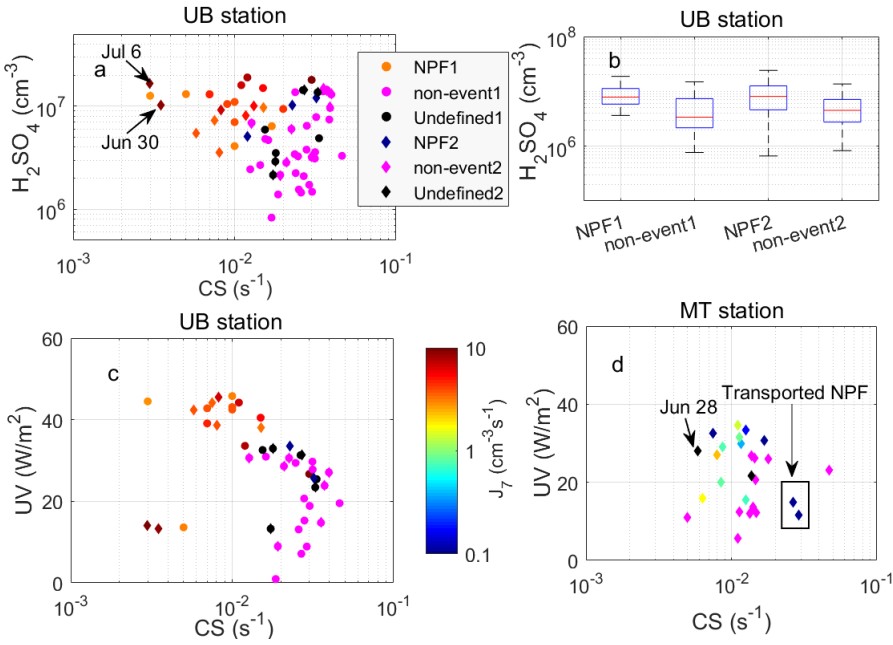

**Figure 9**: (a) Median condensation sinks (CS, s$^{-1}$) and $H_2SO_4$ concentration (SA, cm$^{-3}$) and (b) (b) medians and percentiles of $H_2SO_4$ concentration observed at UB station during the first 2 hours of NPF events and 9:00-11:00 on non-event days. (c) solar radiation (UVA+UVB, W/m$^2$) during the first 2 hours of every NPF event and 9:00-11:00 on every non-event day at UB station. The 'NPF1' and 'non-event1' referred to NPF event and non-event days in summer 2018 and 2019 and the 'NPF2' and 'non-event2' referred to NPF event and non-event days during the observation from June 14 to July 14, 2019.   (d) Median condensation sinks (CS, s$^{-1}$) and solar radiation (UVA+UVB, W/m$^2$) during the first 2 hours of every NPF event and 9:00-11:00 on every non-event day at MT station. Transported NPF event cases and one non-event day with air masses belonging to west group (Jun 28) were all pointed out in the figure. Size of data points on NPF event days means particle formation rate ($J_7$, cm$^{-3}$s$^{-1}$) when it can be calculated reliably. The time resolution of CS was 8 min at UB station and 4 min at MT station, respectively. The time resolution was 30 min for SA data at UB station and 1h for solar radiation data at both stations.

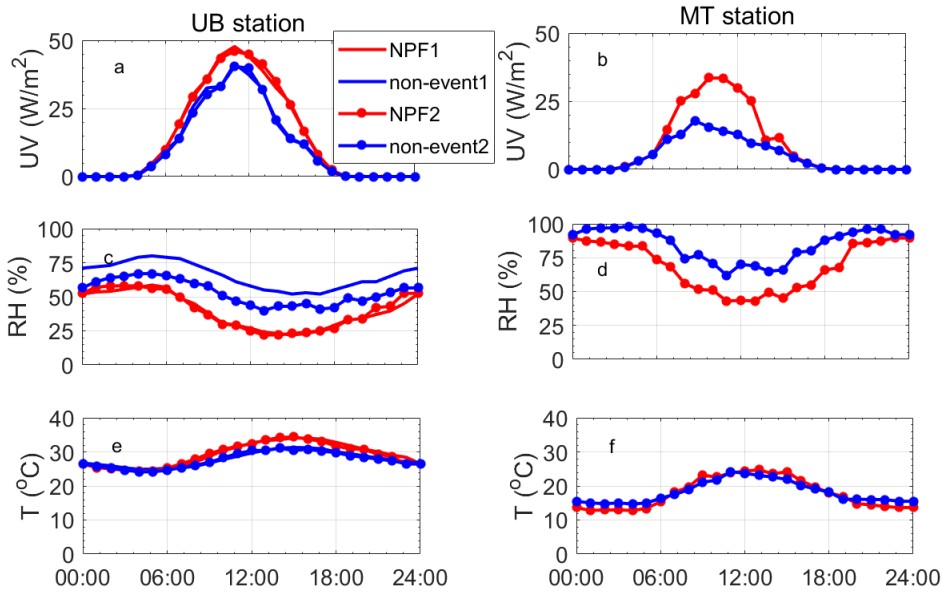

**Figure 10:** (a, b) Diurnal pattern of solar radiation (UV, W/m$^2$), (c, d) Temperature (T, ℃), and (e, f) Relative humidity (RH, %), at UB (left panel) and MT (right panel) stations on both NPF event and non-event days. Time resolutions for all data points here were 1h. The 'NPF1' and 'non-event1' referred to NPF event and non-event days in summer 2018 and 2019 and the 'NPF2' and 'non-event2' referred to NPF event and non-event days during the observation from June 14 to July 14, 2019.

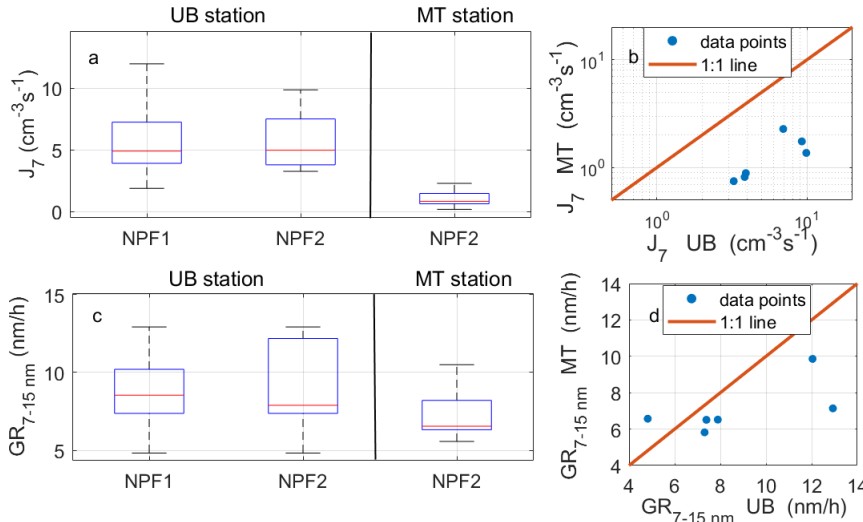

1035

**Figure 11:** Median and percentages of particle formation rates of 7 nm ($J_7$, $cm^{-3}s^{-1}$) (a) and particle growth rates from 7 to 15 nm ($GR_{6-15\ nm}$, nm/h) (c) measured at both stations during our observation as well as comparison between $J_7$ (b) and $GR_{6-15\ nm}$ (d) of common NPF events. The red line represents the median of the data and the lower and upper edges of the box represent $25^{th}$ and $75^{th}$ percentiles of the data, respectively. The length of the whiskers represents 1.5× interquartile range which includes 99.3% of the data. The 'NPF1' and 'non-event1' referred to NPF event and non-event days in summer 2018 and 2019 and the 'NPF2' and 'non-event2' referred to NPF event and non-event days during the observation from June 14 to July 14, 2019.


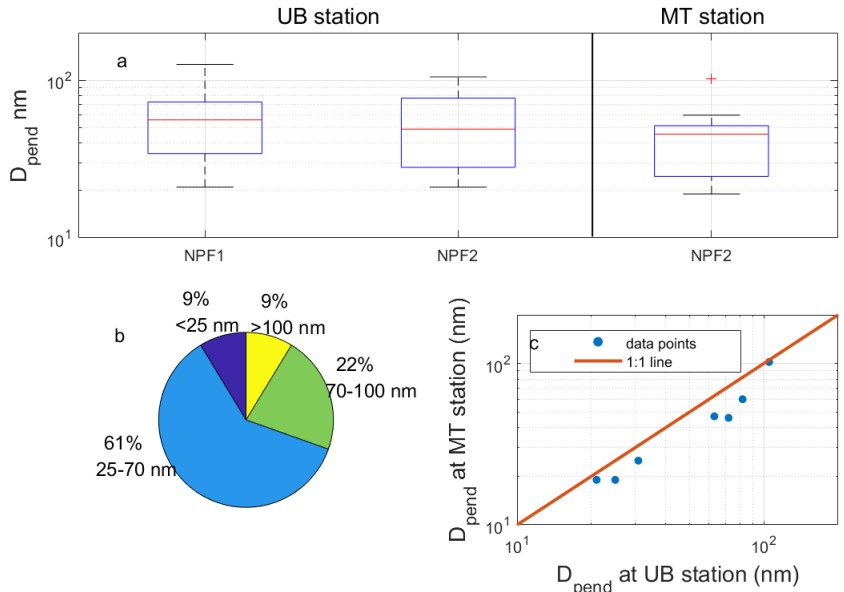


Figure 12: (a) Median and percentiles of end diameters ($D_{pend}$, nm) of NPF events measured
at both sites. The red line represents the median of the data and the lower and upper edges of
the box represent 25[th] and 75[th] percentiles of the data, respectively. The length of the whiskers
represents 1.5× interquartile range which includes 99.3% of the data. The 'NPF1' and
'non-event1' referred to NPF event and non-event days in summer 2018 and 2019 and the
'NPF2' and 'non-event2' referred to NPF event and non-event days during the observation
from June 14 to July 14, 2019. (b) Frequencies of end diameters in the size range of smaller
than 25 nm, 25-70 nm, 70-100 nm and above 100 nm during our observation at UB station in
summer 2018 and 2019. (c) Comparison between end diameters of common NPF events at
both stations.

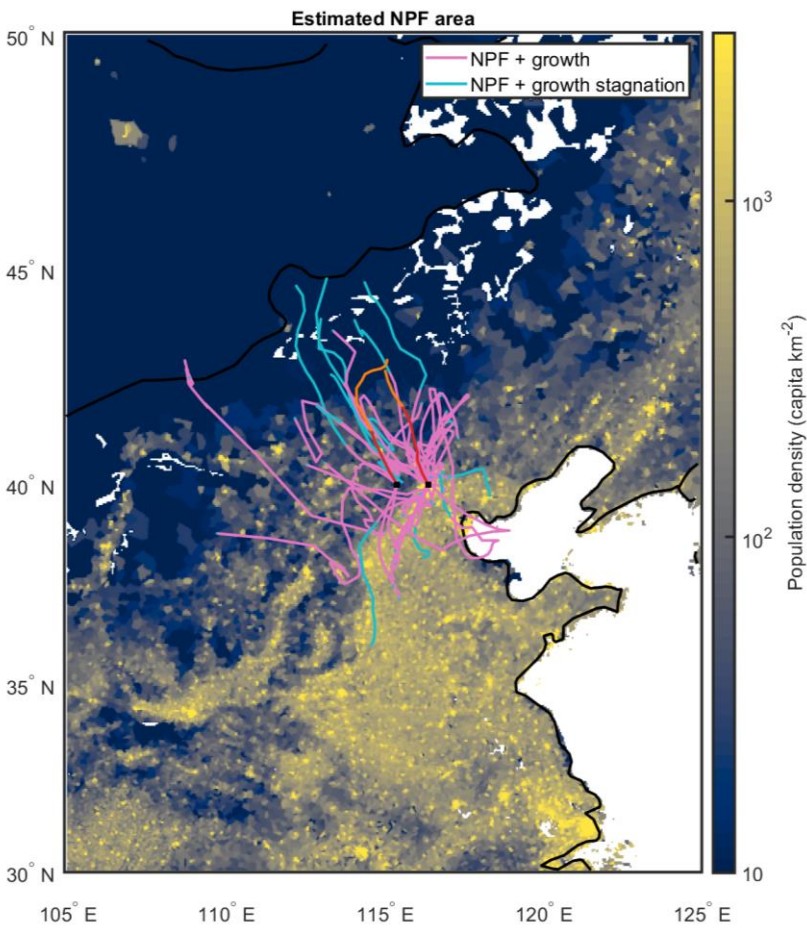


**Figure 13:** Spatial extent of the area where new particle formation events are estimated to have taken place based on air mass back trajectories and the observed NPF events at both sites. Each line represents a single NPF event and extends to the point beyond which continuation of the mode formed in an NPF event was no longer observed at the measurement site. In other words, if an air mass is located outside the area roughly outlined by the colored lines during the typical onset time of NPF and then transported to our measurement sites, NPF is unlikely to have occurred in said air mass. The lines change color from pink to light blue if the observed NPF event enters a stage of growth stagnation, which can indicate a less favorable environment for the formation and growth of new particles. The lines for the case study day of June 30, 2019 are marked with red and change color to orange if growth stagnation occurs. The lines are overlaid on top of a population density map (Gridded Population of the World; GPWv4.10; CC BY 4.0), which is used to illustrate the level of anthropogenic activities and emissions.

1072

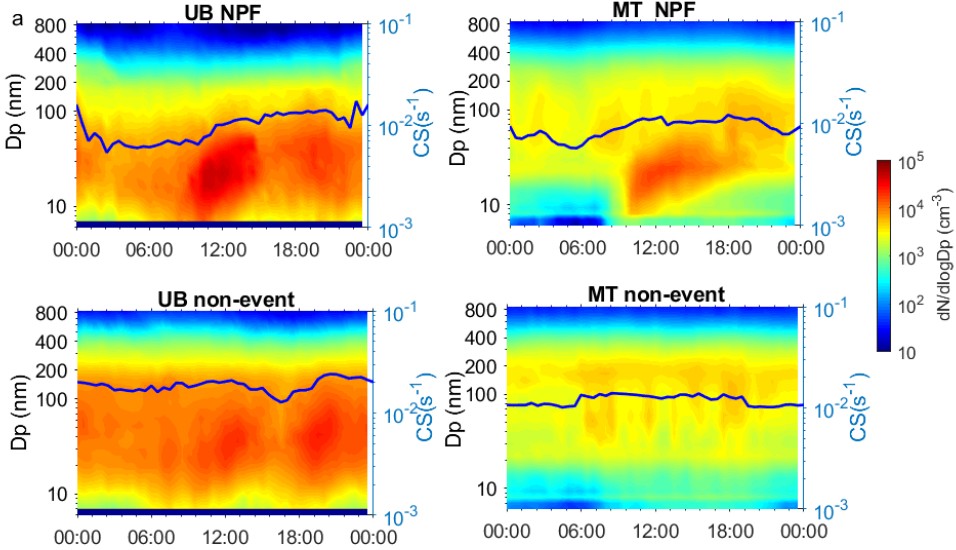

1073

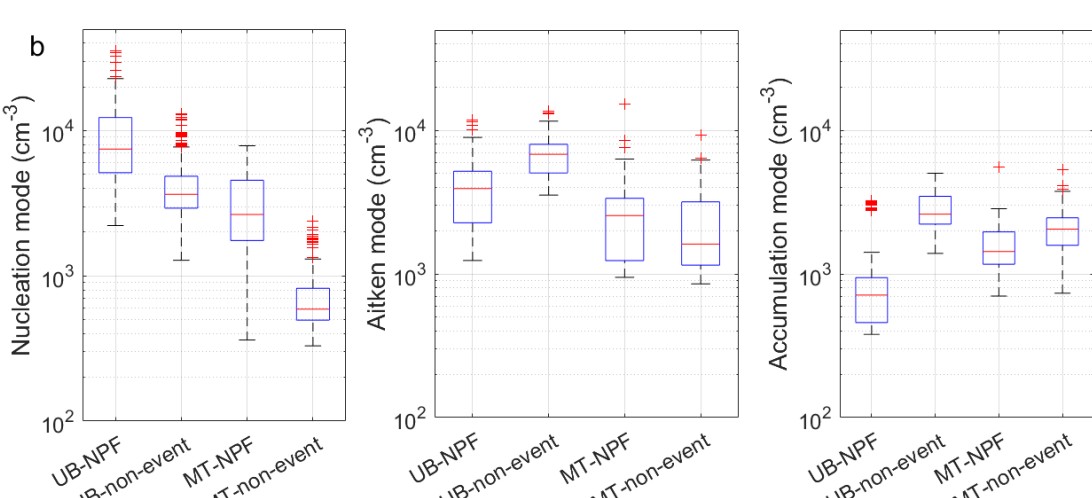

1074

**Figure 14:** (a) Median particle number size distribution as well as CS (blue lines) on NPF event and non-event days at UB (left panel) and MT (right panel) stations and (b) median and percentiles of nucleation, Aitken and accumulation modes particle number concentration on NPF event and non-event days during our observation from June 14 to July 14, 2019 at both stations. The red line represents the median of the data and the lower and upper edges of the box represent 25th and 75th percentiles of the data, respectively. The length of the whiskers represents 1.5× interquartile range which includes 99.3% of the data. Data outside the whiskers are considered outliers and are marked with red crosses.



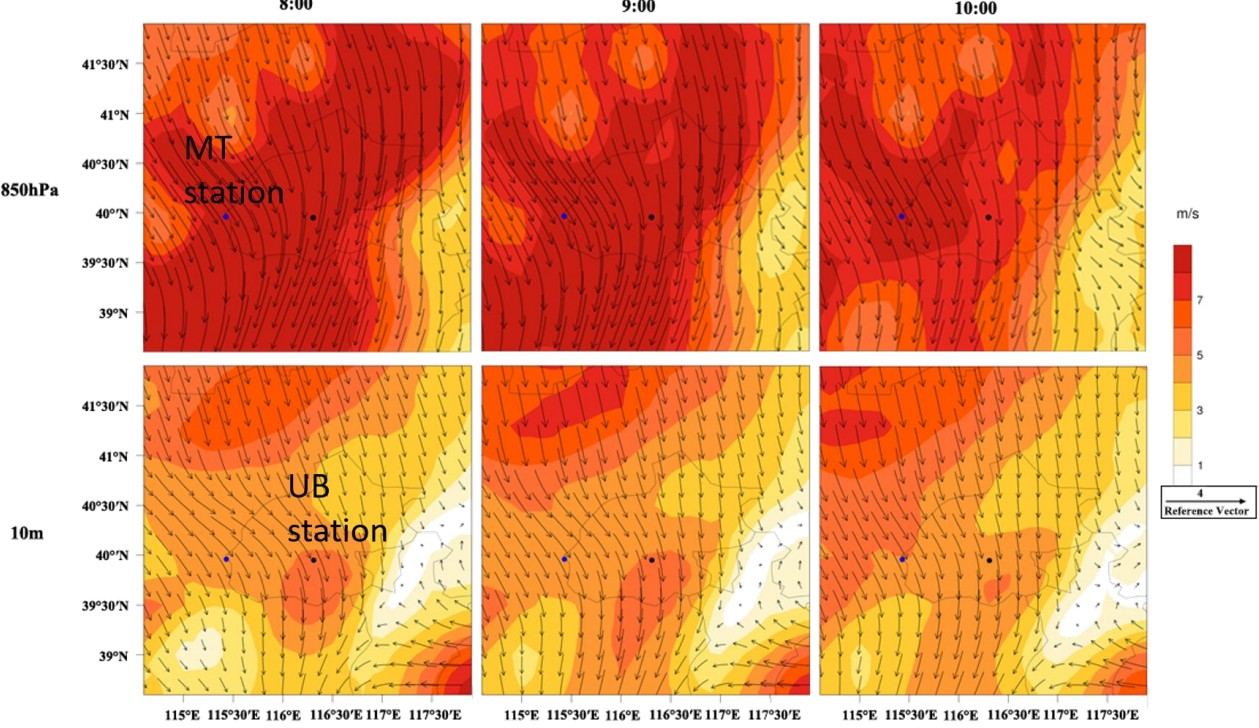


Figure 15: Wind distribution at 8:00, 9:00 and 10:00 on June 30, 2019 at 10 m above the
ground level (lower panel) and 850 hPa (close to the altitude of MT station, upper panel). The
blue and black points on the figures represent MT and UB stations, respectively.

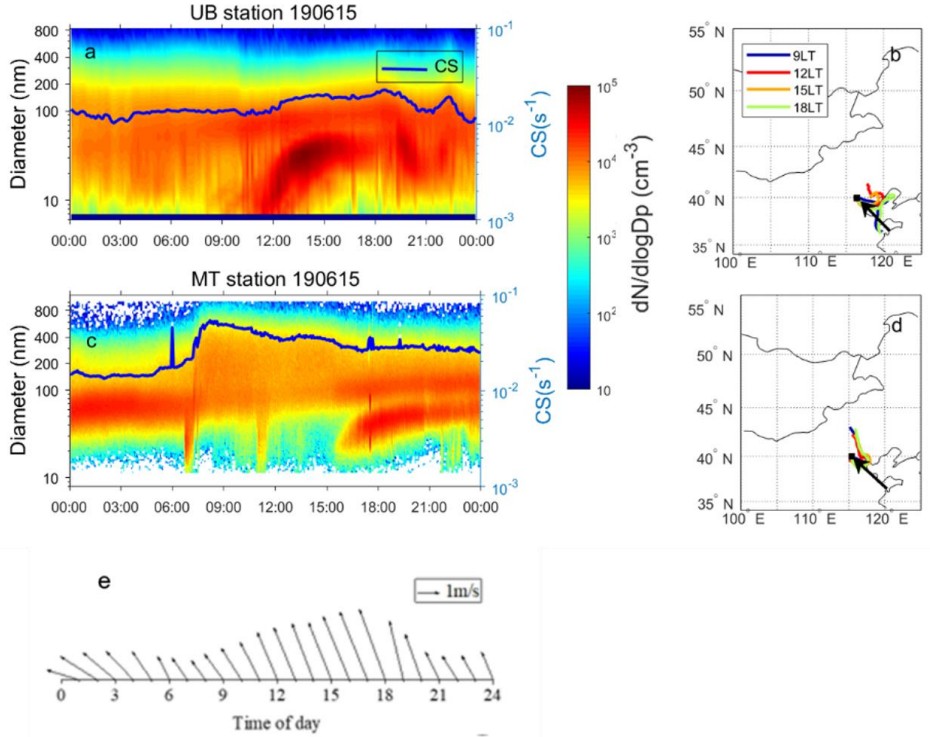


**Figure 16:** Time series of particle number size distribution, CS (blue lines) and air masses arrived at UB (upper panel) and MT (bottom panel) stations as well as wind conditions at MT station on June 15, 2019. Time resolution for particle number size distribution data and CS were both 8 min at UB station and 4 min at MT station, respectively. Time resolution for wind condition data was 1h at MT station. The arrows in the figure denotes directions of prevailing air masses before arriving at both stations during 9:00-15:00 LT.

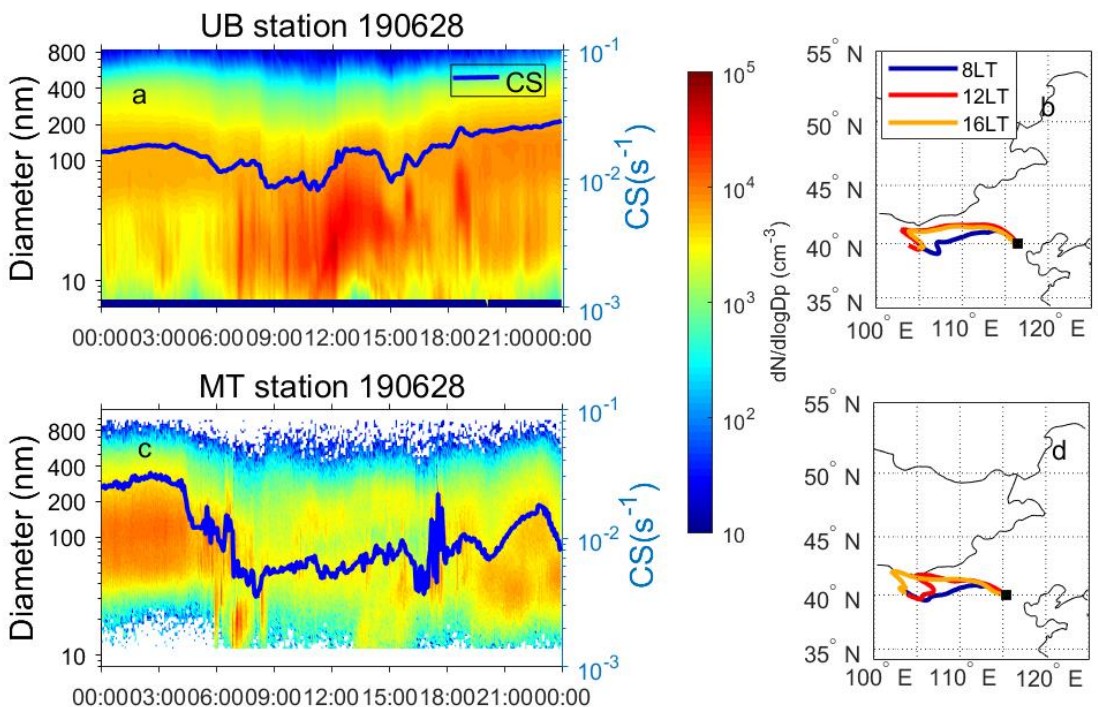

1097

**Figure 17:** Time series of particle number size distribution, CS and air masses arrived at UB

(upper panel) and MT (bottom panel) stations on June 28, 2019. Time resolution for particle

number size distribution data and CS were both 8 min at UB station and 4 min at MT station,

respectively.



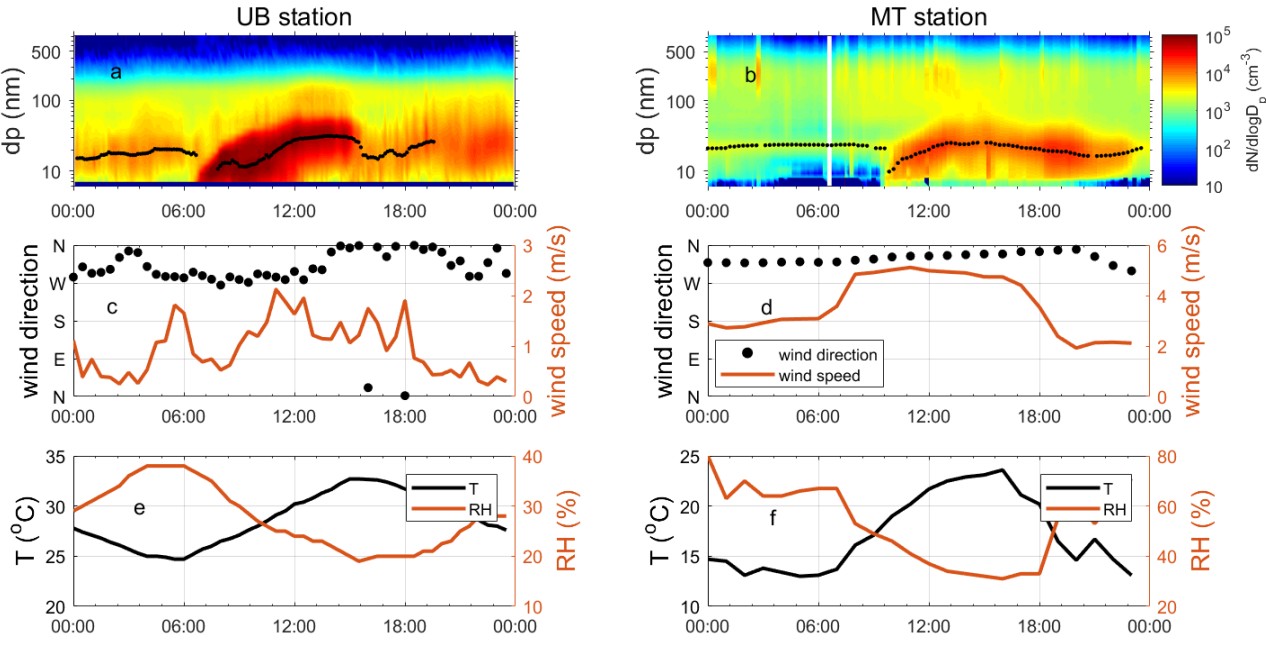


Figure 18: Time series of particle number size distribution and mode diameters (a, b), wind
speed and direction (c, d), temperature and RH (e, f) measured at UB (left panel) and MT
(right panel) on June 30, 2019.