# Peer review of "Measurement Report: New particle formation characteristics"

_Atmospheric Chemistry and Physics, 2021_

## Referee Comment (RC1)

NPF is a hot atmospheric topic and its air quality, climate and human health effect still remains not clear. As a measurement report, only 25 days are available, which can not provide robust statistical results of NPF parameters (frequency, GR and FR). The author compared the NPF parameters with the previous studies in NCP region, however, the short period study can't explain the difference, but cause large uncertainties in statistical values. In this work, something new about the NPF events study should be pointed out as many similar studies have been conducted in the same region in China.

Major concerns:
1. Statistical significance: The statistical numbers of formation rate, growth rate, etc. was calculated based on only a few NPF cases (12, 13 NPF events at different locations). Is this statistical meaningful as the small quantity of cases? Also the comparison of the NPF frequency, GR, FR, and CS with the previous long-term study should be careful.
2. Instrument consistency: in the section 2.2, there is DMPS, SMPS, FMPS used in the PNSD measurement, the comparison of PNSD derived by different instrument should be given of the overlap size range. It is very important to make sure the data are comparable, as the PNSD data also determine the formation rate, growth rate and CS. The type and manufacturer of DMA of DMPS, as well as the SMPS should be also provide.
3. The influence of air mass origin on the regional NPF occurrence was discussed. As the MT site locates nearby the mountain, how does the topography affect the air mass, local wind, as well as the inhomogeneity of regional NPF events should be also addressed.

Minor comments:
1. L18-19, I don't think this conclusion is appropriate in the abstract, as the sentence imply this is the first work about urban and regional measurement. Actually, Wang et al., (2013) has reported the regional NPF events in urban Beijing and a regional background site based on one-year dataset before.
2. L47, the reference of the same author should be cited as Guo et al., 2014; 2020.
3. L217, what is "good data"? The clear explanation should be given.
4. L222, higher NPF frequency in this study, as compared with Wang et al., 2013; Deng et al., 2020a, was explained by "25 days were validated data", it is not convincing. The short period study caused large uncertainty in the comparison, including the formation rate, growth rate, frequency and CS. However, it can't explain for the higher or lower value. Other favorable parameters for NPF, e.g. meteorology, precursors, CS, should also be taken into consideration.
5. Section 3.1.4, do you mean the higher ending diameter at UB site, supported by the higher condensing vapors? But as you have mentioned, the GR at both sites were comparable, does that mean the condensing level are also comparable? The conclusions from GR and ending diameter were not consistent, it should be discussed further.
6. Figure 1: why the station of S60 is shown in the figure?
7. The key word: haze, there is no much discussion about how NPF event contributing

to regional haze formation in the study. So I think this key word is not appropriate.

---

## Referee Comment (RC2)

*Comments:*

This work reported simultaneous measurements of new particle formation at an urban and a mountain station in Northern China. The manuscript fits well to the scope of ACP, however, I do not think it represented new results or findings. The paper is worth to be published, but not in its current form. Thus I recommend it to be resubmitted if more analysis or dataset could be included.

Lots of paper about NPF in North China Plain have been published. I would suggest the authors demonstrate the new findings compared to the previous studies. As a measurement report, it only present one month data, I think this is enough. The conclusions did not show anything new.

The authors pointed out that NPF events at urban areas could have a bigger influence on global/regional climate and air pollution than those at clean areas. This should be considered in regional-scale aerosol models when estimating the budget of aerosol and CCN loadings. I agree with this, but how could we deal with it in the model. Maybe some exemplary cases and detailed discussions are more convincing.

to regional haze formation in the study. So I think this key word is not appropriate.

**References**

---

## Author Comment (AC1)

*Measurement Report: New particle formation characteristics at an urban and a mountain station in Northern China by Ying Zhou et al.*

In this file, the referee comments are in black, our item-by-item replies are in blue, and the corresponding modifications in the manuscript are in red.

**Answers to reviewer # 1**

NPF is a hot atmospheric topic and its air quality, climate and human health effect still remains not clear. As a measurement report, only 25 days are available, which cannot provide robust statistical results of NPF parameters (frequency, GR and FR). The author compared the NPF parameters with the previous studies in NCP region, however, the short period study can't explain the difference, but cause large uncertainties in statistical values. In this work, something new about the NPF events study should be pointed out as many similar studies have been conducted in the same region in China.

As per suggestion of the referee, we extended our data set at urban Beijing (UB) to include two full summers (from June 1 to August 31) in both 2018 and 2019. With the extended data, we updated our discussion on favorable and limiting factors for NPF event occurrence at the urban Beijing station. We also estimated the size of the area within which regional NPF events occurred during our observations. This estimation could possibly explain the phenomenon that most ending diameters of NPF events during our observation were limited to below 70 nm. Our results highlight the importance of anthropogenic emissions for NPF occurrence and subsequent growth in summer Beijing. Unfortunately, we were not able to include more data from the background mountain station (MT), since no long-term measurements were maintained. Yet, we are able to observe the contribution of the MT in forming particles and its difference from the UB station. Our study highlights the importance of establishing and maintaining long-term measurement not only in urban locations but also in rural ones.

Major concerns:

1. Statistical significance: The statistical numbers of formation rate, growth rate, etc. was calculated based on only a few NPF cases (12, 13 NPF events at different locations). Is this statistical meaningful as the small quantity of cases? Also the comparison of the NPF frequency, GR, FR, and CS with the previous long-term study should be careful.

As mentioned above and as per suggestion of the referee, we extended our data set at urban Beijing station (UB) to include two full summers (from June 1 to August 31) in 2018 and 2019 to improve our statistical conclusions. With the extended data, we found that the shorter observation period between June 14 to July 14, 2019 is representative of the urban Beijing summer as no significant difference in NPF events' properties were observed when comparing the observation from June 14 to July 14, 2019 to the two full summer periods in 2018 and 2019. We then updated the comparison of particle growth rate, formation rate and CS based on the data measured in summer 2018 and 2019 at UB station.

*The following discussion and figures (Figure R1-1, Figure R1-2 and Figure R1-3) will be added to the manuscript:*

Figure R1-1a shows the difference in CS between NPF event and non-event days during our observation in summer 2018 and 2019(two whole summers)at UB site and short-term parallel observations at both sites. The 'NPF1' and 'non-event1' referred to NPF and non-event days during the two whole summers, respectively, while 'NPF2' and 'non-event2' referred to NPF and non-event days during the short-term parallel observation period from June 14 to July 14, 2019 at both sites, respectively. The longer-term periods are used for confirming the representativeness of the short-term overlapping period for the whole summer. As shown in the figure, the median CS on NPF1 or NPF2 days is equivalent for UB station ($CS_{NPF1}$ = 0.010$s^{-1}$; $CS_{NPF2}$= 0.009$s^{-1}$) and less than a factor of 1.2 different between non-event1 and non-event2 in UB station ($CS_{nonevent1}$ = 0.023$s^{-1}$; $CS_{nonevent2}$= 0.020$s^{-1}$), which confirms the representativeness of our short-term measurement period to the overall urban Beijing summer.

Our results in Fig. R1-1a show that on NPF event days, the median CS was ~ 0.01 $s^{-1}$ during the first 2 hours of the NPF events, at both stations. On common NPF event days, the median

CS was 0.009 s$^{-1}$ at UB station and ~0.01s$^{-1}$ at MT station, respectively. In comparison, on non-event days, during roughly the same time period (9:00–11:00 LT), the CS was substantially higher, with median values of 0.02 s$^{-1}$ and 0.014 s$^{-1}$, at UB and MT stations, respectively. Figure R1-1b presents the median CS during the first 2 hours of NPF events on common NPF event days measured at both stations, and shows the high correlation between the two.

Figure R1-1c shows the NPF event frequency as a function of CS during our observation at UB site in summer 2018 and 2019 and how the NPF event frequency decreased with increasing CS. When CS was smaller than 0.01 s$^{-1}$, all days were classified as NPF event days, and when CS was larger than 0.035 s$^{-1}$, no day was classified as NPF event day. This shows the major role of background particles in controlling the occurrence or inhibition of NPF events as shown in several previous studies in China and internationally (Deng et al., 2020a; Cai et al., 2017; Kulmala et al., 2017). While we cannot present a similar figure from the MT station, the same conclusion applies where CS does play a role in inhibiting NPF observation owing to the difference in the CS values observed between NPF and nonevents at MT in Fig. R1-1a. Yet, since the overall preexisting particle concentration at the MT is rather on the low end, the role of CS might not be as vital at the MT station as for the UB station.

The particle formation rates ($J_7$) at the two stations during the measurements are presented in Fig.R1-2a. $J_7$ observed during the short-term parallel observation (NPF2) at UB site was in the range of 3.0-10.0 cm$^{-3}$ s$^{-1}$ with a median of 5.4 cm$^{-3}$ s$^{-1}$, comparable with those observed in summer 2018 and 2019 (NPF1 = 2-14.0 cm$^{-3}$ s$^{-1}$ with a median of 4.9 cm$^{-3}$ s$^{-1}$) and significantly higher than the values in the MT station (0.75-3.0 cm$^{-3}$ s$^{-1}$ with a median of 0.82 cm$^{-3}$s$^{-1}$) for common NPF events (Fig.R1-2b).

The particle growth rates in size range of 7-15 nm (GR$_{7-15nm}$) at the UB station (4.8-12.9 nm/h with a median of 7.8 nm/h) during NPF2 was also comparable with the whole summers (NPF1) (4.8-12.9 nm/h with a median of 8.5 nm/h). While the difference in $J_7$ was 7 times higher in UB than in MT, the observed GR were only slightly higher at UB than at the MT station (5.7-10.5 nm/h with a median of 6.5 nm/h) for common NPF events (Fig. R1-2c&d), implying that precursors needed for particle formation were much more abundant in the

polluted urban environment (Wang et al., 2013), while those needed for growth are rather comparable.

2. Instrument consistency: in the section 2.2, there is DMPS, SMPS, FMPS used in the PNSD measurement, the comparison of PNSD derived by different instrument should be given of the overlap size range. It is very important to make sure the data are comparable, as the PNSD data also determine the formation rate, growth rate and CS. The type and manufacturer of DMA of DMPS, as well as the SMPS should be also provide.

We thank the reviewer for the suggestions.

*The following discussion and figures (Figure R2-1) will be added to the manuscript in section 2.2:*

The DMPS consists of consists of one Hauke-type DMA (differential mobility analyzer, home-built by university of Helsinki) in different flow rates and one CPC (condensation particle counter, TSI Model 3772). Details of this instrument can be found in Salma et al., (2011) and Kangasluoma et al. (2020).

At MT station, a scanning mobility particle sizer (SMPS, consists of a TSI Differential Mobility Analyzer model 3081) and a fast mobility particle sizer (FMPS, TSI Model 3091) were used to measure particle number size distribution from June 14 to June 28 and from June 29 to July 14, respectively. The particle number size distribution measured by FMPS correlated well with SMPS after being calibrated (Lee et al., 2013).

To ensure high quality of particle number size distribution data at UB site, a particle number size distribution system (PSD) also sampled in parallel with DMPS from June 1 to August 31, 2019 (summer 2019). It measured particle number size distribution in the size range of 1 nm to 10 µm. It included a nano-scanning mobility particle sizer (nano-SMPS, 3–55 nm, mobility diameter), a long SMPS (25–650 nm, mobility diameter) and an aerodynamic particle sizer (APS, 0.55–10 µm, aerodynamic diameter). Details of this instrument can be seen at Liu et al. (2016) and Deng et al. (2020b).

As shown in Fig. R2-1, median particle number size distribution obtained from PSD and DMPS matched well with each other within a factor of 2 during our observation in summer

2018 and 2019 at UB site. We cannot compare particle number size distribution data obtained from DMPS, SMPS and FMPS as we did not sample with these three instruments in parallel at the same site. However, it is reasonable to assume that particle number size distribution obtained from FMPS were comparable with those from DMPS as on one hand the measurement techniques of particle number size distribution in the size range of these two instruments have been well developed and are applied in quite a lot observations in several environments (Wang et al., 2017;Kangasluoma et al., 2020). On the other hand, the FMPS was carefully calibrated and properly operated during the observation as discussed above. Similar conclusions apply for the SMPS as well, where we can rely on using the measurement from this instrument to discuss at least NPF event frequency at MT site during June 14 to June 28, 2019, during which parameters of only one NPF event are calculated.

3. The influence of air mass origin on the regional NPF occurrence was discussed. As the MT site locates nearby the mountain, how does the topography affect the air mass, local wind, as well as the inhomogeneity of regional NPF events should be also addressed.

*We thank the referee for the suggestions. As per suggestions of the referee, the following discussions and figures (Figure R3-1 and Figure R3-2) are added to the manuscript results section:*

**3.5 Effect of topography**

In Figure R3-1 we show average particle number size distribution and particle number concentration on NPF event and non-event days during our short-term parallel observation at both sites. On NPF event days, nucleation and Aitken mode particle number concentrations were much smaller at MT station than those at UB station due to smaller particle formation rates and less anthropogenic emissions. Interestingly, accumulation mode particle number concentrations were higher at MT station (701-2900 cm$^{-3}$, with a median of 1500 cm$^{-3}$) than that at UB station (350-1416 cm$^{-3}$ with a median of 700 cm$^{-3}$) (Fig.R3-1b). Due to the close proximity of the two measurement sites, the air mass arrival directions and source regions were (mostly) similar at both sites throughout the short-term parallel measurement period (Table 1) hence the regional and transported cannot explain the higher accumulation mode

particle number concentration at MT site. As there were few primary emissions at MT site, the accumulation mode particles could be attributed to secondary particles (Kulmala et al., 2021), indicating particles at MT station were more aged than those at UB station (Fig.R3-1a). The possible reason is that mountains block pollution diffusion, which in the end resulted in comparable CS at MT station as UB station.

Figure R3-2 shows an example of the wind distribution before and during NPF event on June 30, 2019 at 850 hPa (close to the altitude of MT station) and 10 m above ground level. As shown in Fig. R3-2, the reanalyzed wind directions at 850 hPa were similar as those at 10 m above the ground level at MT station. Actually, the wind conditions on other NPF event days at MT station during our observation had similar characteristics that the wind directions were similar between 850 hPa and 10 m above ground level indicating air masses well mixed during NPF events. Earlier observations also found NPF event happened uniformly within the mixing layer at their observation stations and particle number size distribution remains roughly constant within the mixing layer (Shen et al., 2018;Lampilahti et al., 2021).

Minor comments:

1. L18-19, I don't think this conclusion is appropriate in the abstract, as the sentence imply this is the first work about urban and regional measurement. Actually, Wang et al., (2013) has reported the regional NPF events in urban Beijing and a regional background site based on one-year dataset before.

We thank the referee for the comments. We agree that our work about urban and regional measurement is not the first but the observations on NPF events on the mountain is still rare in China emphasizing the need of establishing and maintaining such long term observations there.

*As per suggestions of the referee, we corrected the sentence in line 18-19 as below:*

Most observations on NPF events in Beijing and its vicinity were conducted in populated areas, whereas observations on NPF events in mountain sites with few anthropogenic emissions are still rare in Beijing (Wang et al., 2013). The spatial variation of NPF event intensity has not been investigated in detail by incorporating both urban area and mountain measurements.

2. L47, the reference of the same author should be cited as Guo et al., 2014; 2020.

We corrected the citation as Guo et al., 2014; 2020 in our manuscript.

3. L217, what is "good data"? The clear explanation should be given.

The "good data" meant that particle number size distribution were valid that visual inspection of the data and the number concentrations as well as instrument status do not indicate problems in the measurements at both sites. As per suggestions to the reviewer, we corrected sentence in line 217 as below:

Only days when particle number size distribution data were valid that visual inspection of the data and the number concentrations as well as instrument status do not indicate problems in the measurements for both stations were taken into consideration in our analysis.

4. L222, higher NPF frequency in this study, as compared with Wang et al., 2013; Deng et al., 2020a, was explained by "25 days were validated data", it is not convincing. The short period study caused large uncertainty in the comparison, including the formation rate, growth rate, frequency and CS. However, it can't explain for the higher or lower value. Other favorable parameters for NPF, e.g. meteorology, precursors, CS, should also be taken into consideration.

We thank the referee for the comments. As per suggestions to the reviewer, the following discussions and figures (Fig.R4-1, Fig.R4-2 and Fig.R4-3) will be added in our manuscript:

In Fig.R4-1, we show frequencies of air masses arriving at UB station from different directions during our observation in summer 2018 and 2019. The most frequent air masses arriving at UB station belonged to the South group. During our observation in the two summers, out of 155 days were 52 days belonging to the South group and 39, 32, 9 and 23 days in air masses belong to North, East, West and Local groups, respectively. NPF event frequency with respect to air masses is also shown in Fig. R4-1. It is noticeable that air mass origin influenced the occurrence of NPF events at UB site as the majority of NPF events occurred when the air masses were coming from the north. During our observation in summer 2018 and 2019, 34 (out of 55) NPF events occurred in air masses from the North group and 9, 2, 2 and 6 NPF events in the South, East, West and Local groups, respectively (Fig.R4-1a). One prominent feature of these air masses is their difference in CS. As shown in Fig. R4-1b,

the CS of the air masses classified as the North group (with median values of 0.01 s$^{-1}$ at UB station) is substantially lower than that in other air mass classes (CS = 0.03, 0.025, 0.017, 0.03 s$^{-1}$, for south, east, west and local, respectively), which might explain the high NPF event frequency associated with this air mass class. During the observation from June 14 to July 14 in summer 2019, the most frequent air masses arriving at both sites belonged to the North group as shown in Table 1. Out of 25 days, there were 8 and 9 days belonging to the North group, at UB and MT sites, respectively. The highest frequency of NPF events also occurred when the air masses were coming from the north. The high NPF events frequency during our observation form June 14 to July 14 could also be attributed to the frequent air masses arriving at both sites from north to Beijing.

In Figure R4-2, we show diurnal variation of meteorological variables during our observation in summer 2018 and 2019 at UB site and observations from June 14 to July 14 in 2019 at UB and MT sites. It is noticeable that the short-term observation compared well with the long-term observation and therefore is representative of summer at UB site as shown in Fig.R4-2.

In Fig. R4-3a, we show the concentration of sulfuric acid as a function of CS during summer 2018 and 2019 at UB site. As shown in Fig. R4-3b, the median sulfuric acid ($H_2SO_4$) concentrations at UB station were $8.1\times10^6$ cm$^{-3}$ and $4.5\times10^6$ cm$^{-3}$ on NPF event days and non-event days, respectively, during observation from June 14 to July 14 in 2019 and $7.9\times10^6$ cm$^{-3}$ and $3.4\times10^6$ cm$^{-3}$ on NPF event days and non-event days, respectively, during the observation in summer 2018 and 2019. This suggests that $H_2SO_4$ was important for NPF events at the UB station (Deng et al., 2020b; Dada et al., 2020b). On the other hand, as shown in Fig.R4-3a, the $H_2SO_4$ concentration during 9:00- 11:00 (local time) on non-event days could be comparable with that on NPF event days, especially when CS was high. Altogether, our observation shows that the occurrence of NPF events was controlled by both $H_2SO_4$ and CS at the UB station (Cai et al., 2020).

5. Section 3.1.4, do you mean the higher ending diameter at UB site, supported by the higher condensing vapors? But as you have mentioned, the GR at both sites were comparable, does that mean the condensing level are also comparable? The conclusions from GR and ending

diameter were not consistent, it should be discussed further.

We thank the referee for the comments. We found that the ending diameters were slightly higher at UB site than at MT site, but the difference is not significant (49 nm vs 45 nm), in addition, the GR on common NPF event days were a little bit higher at UB site than MT site. Earlier research pointed out that in order to observe the growth until 100 nm at the measurement station under typical conditions, simultaneous NPF should happen in a very large area (e.g. with wind speed 5m/s and growth rate of 3nm/h from the station to roughly 600 km upwind from the station) (Paasonen et al., 2018). To discuss the ending diameters, we evaluated the size of regions that NPF events could be occurring during our observation. Also, as per suggestions of the reviewer, we updated our comparison of GR on common NPF events as well as the discussion on ending diameters as follows:

The particle growth rates in size range of 7-15 nm ($GR_{7-15nm}$) at the UB station (4.8-12.9 nm/h with a median of 7.8 nm/h) during observation from June 14 to July 14, 2019 was also comparable with that during the observation in summer 2018 and 2019, a little bit higher than that in the MT station (5.7-10.5 nm/h with a median of 6.5 nm/h) for common NPF events (Fig.R1-2c&d), implying that precursors were only slightly more abundant in the polluted urban environment (Wang et al., 2013).

*Addition to the methods' section:*

**2.4 Estimating the spatial extent of NPF**

[revised manuscript text omitted]

6. Figure 1: why the station of S60 is shown in the figure?

We thank the referee for the comments. The S60 referred to the location where particles formed during the non-local NPF event observed at MT station on June 15, 2019.

We added the following sentence in our manuscript:

**Figure 1:** Map showing locations of urban station (UB), Longquan station (LQ), mountain station (MT) and another site 60 km south from MT station (S60). The S60 referred to the location where particles formed during the non-local NPF event observed at MT station on June 15, 2019. Image is produced using © Google Maps.

7. The key word: haze, there is no much discussion about how NPF event contributing to regional haze formation in the study. So I think this key word is not appropriate.

We thank the referee for the comments. As per suggestion to the reviewer, we removed the key word "haze".

**Figures**

[revised manuscript text omitted]

Kulmala, M., Dada, L., Dällenbach, K., Yan, C., Stolzenburg, D., Kontkanen, J., Ezhova, E., Hakala, S., Tuovinen, S., Kokkonen, T., Kurppa, M., Cai, R., Zhou, Y., Yin, R., Baalbaki, R., Chan, T., Chu, B., Deng, C., Fu, Y., Ge, M., He, H., Heikkinen, L., Junninen, H., Nei, W., Rusanen, A., Vakkari, V., Wang, Y., Wang, L., yao, l., Zheng, J., Kujansuu, J., Kangasluoma, J., Petäjä, T., Paasonen, P., Järvi, L., Worsnop, D., Ding, A., Liu, Y., Jiang, J., Bianchi, F., Yang, G., Liu, Y., Lu, Y., and Kerminen, V.-M.: Is reducing new particle formation a plausible solution to mitigate particulate air pollution in Beijing and other Chinese megacities?, Faraday Discuss, 10.1039/d0fd00078g, 2021.

Liu, J. Q., Jiang, J. K., Zhang, Q., Deng, J. G., and Hao, J. M.: A spectrometer for measuring particle size distributions in the range of 3 nm to 10 mu m, Front Env Sci Eng, 10, 63-72, https://doi.org/10.1007/s11783-014-0754-x, 2016.

Lee, B. P., Li, Y. J., Flagan, R. C., Lo, C., and Chan, C. K.: Sizing characterization of the fast mobility particle sizer (FMPS) against SMPS and HR-ToF-AMS, Aerosol Sci. Technol., 47, 1030–1037, https://doi.org/10.1080/02786826.2013.810809, 2013.

Liu, J. Q., Jiang, J. K., Zhang, Q., Deng, J. G., and Hao, J. M.: A spectrometer for measuring particle size distributions in the range of 3 nm to 10 mu m, Front Env Sci Eng, 10, 63-72, https://doi.org/10.1007/s11783-014-0754-x, 2016.

Wang, Z. B., Hu, M., Sun, J. Y., Wu, Z. J., Yue, D. L., Shen, X. J., Zhang, Y. M., Pei, X. Y., Cheng, Y. F., and Wiedensohler, A.: Characteristics of regional new particle formation in urban and regional background environments in the North China Plain, Atmos Chem Phys, 13, 12495-12506, 10.5194/acp-13-12495-2013, 2013.

Ma, L., Zhu, Y., Zheng, M., Sun, Y., Huang, L., Liu, X., Gao, Y., Shen, Y., Gao, H., and Yao, X.: Investigating three patterns of new particles growing to the size of cloud condensation nuclei in Beijing's urban atmosphere, Atmos Chem Phys, 21, 183-200,

10.5194/acp-21-183-2021, 2021.

Man, H. Y., Zhu, Y. J., Ji, F., Yao, X. H., Lau, N. T., Li, Y. J., Lee, B. P., and Chan, C. K.: Comparison of Daytime and Nighttime New Particle Growth at the HKUST Supersite in Hong Kong, Environ Sci Technol, 49, 7170-7178, 2015.

Shen, X., Sun, J., Kivekäs, N., Kristensson, A., Zhang, X., Zhang, Y., Zhang, L., Fan, R., Qi, X., Ma, Q., and Zhou, H.: Spatial distribution and occurrence probability of regional new particle formation events in eastern China, Atmos Chem Phys, 18, 587-599, 10.5194/acp-18-587-2018, 2018.

Lampilahti, J., Manninen, H. E., Nieminen, T., Mirme, S., Ehn, M., Pullinen, I., Leino, K., Schobesberger, S., Kangasluoma, J., Kontkanen, J., Järvinen, E., Väänänen, R., Yli-Juuti, T., Krejci, R., Lehtipalo, K., Levula, J., Mirme, A., Decesari, S., Tillmann, R., Worsnop, D. R., Rohrer, F., Kiendler-Scharr, A., Petäjä, T., Kerminen, V.-M., Mentel, T. F., and Kulmala, M.: Zeppelin-led study on the onset of new particle formation in the planetary boundary layer, Atmospheric Chemistry and Physics Discussions, 10.5194/acp-2021-282, 2021.

Paasonen, P., Peltola, M., Kontkanen, J., Junninen, H., Kerminen, V.-M., and Kulmala, M.: Comprehensive analysis of particle growth rates from nucleation mode to cloud condensation nuclei in boreal forest, Atmos Chem Phys, 18, 12085-12103, 10.5194/acp-18-12085-2018, 2018.

Salma, I., Borsós, T., Weidinger, T., Aalto, P., Hussein, T., Dal Maso, M., and Kulmala, M.: Production, growth and properties of ultrafine atmospheric aerosol particles in an urban environment. Atmospheric Chemistry and Physics. 11. 10.5194/acp-11-1339-2011, 2011.

Zhang, J., Chen, Z., Lu, Y., Gui, H., Liu, J., Wang, J., Yu, T., and Cheng, Y.: Observations of New Particle Formation, Subsequent Growth and Shrinkage during Summertime in Beijing, Aerosol Air Qual Res, 16, 1591-1602, 10.4209/aaqr.2015.07.0480, 2016.

---

## Author Comment (AC2)

*Measurement Report: New particle formation characteristics at an urban and a mountain station in Northern China by Ying Zhou et al.*

In this file, the referee comments are in black, our item-by-item replies are in blue, and the corresponding modifications in the manuscript are in red.

**Answers to reviewer # 2**

This work reported simultaneous measurements of new particle formation at an urban and mountain station in Northern China. The manuscript fits well to the scope of ACP, however, I do not think it represented new results or findings. The paper is worth to be published, but not in its current form. Thus I recommend it to be resubmitted if more analysis or dataset could be included.

Lots of paper about NPF in North China Plain have been published. I would suggest the authors demonstrate the new findings compared to the previous studies. As a measurement report, it only present one month data, I think this is enough. The conclusions did not show anything new.

The authors pointed out that NPF events at urban areas could have a bigger influence on global/regional climate and air pollution than those at clean areas. This should be considered in regional-scale aerosol models when estimating the budget of aerosol and CCN loadings. I agree with this, but how could we deal with it in the model. Maybe some exemplary cases and detailed discussions are more convincing.

As per suggestion of the referee, we extend our data set at urban Beijing (UB) to include two full summers (from June 1 to August 31) in 2018 and 2019. With the extended data, we updated our discussion on favorable and limiting factors on NPF event occurrence at the urban Beijing station. We also estimated the size of the area within which regional NPF events occurred during our observations. This estimation could possibly explain the phenomena that most ending diameters of NPF events during our observation were limited below 70 nm. Our results highlight the importance of anthropogenic emissions for NPF occurrence and subsequent growth in summer Beijing. Unfortunately, we were not able to

include more data from the background mountain station (MT), since no long-term measurements were maintained. As the results of this study, show also the contribution of the MT in forming particles, our study highlights the importance of establishing and maintaining long term measurement not only in urban location but also in rural ones.

*Addition to the methods' section:*

[revised manuscript text omitted]

https://doi.org/10.1039/C6FD00257A, 2017.

Kulmala, M., Dada, L., Dällenbach, K., Yan, C., Stolzenburg, D., Kontkanen, J., Ezhova, E., Hakala, S., Tuovinen, S., Kokkonen, T., Kurppa, M., Cai, R., Zhou, Y., Yin, R., Baalbaki, R., Chan, T., Chu, B., Deng, C., Fu, Y., Ge, M., He, H., Heikkinen, L., Junninen, H., Nei, W., Rusanen, A., Vakkari, V., Wang, Y., Wang, L., yao, l., Zheng, J., Kujansuu, J., Kangasluoma, J., Petäjä, T., Paasonen, P., Järvi, L., Worsnop, D., Ding, A., Liu, Y., Jiang, J., Bianchi, F., Yang, G., Liu, Y., Lu, Y., and Kerminen, V.-M.: Is reducing new particle formation a plausible solution to mitigate particulate air pollution in Beijing and other Chinese megacities?, Faraday Discuss, 10.1039/d0fd00078g, 2021.

Liu, J. Q., Jiang, J. K., Zhang, Q., Deng, J. G., and Hao, J. M.: A spectrometer for measuring particle size distributions in the range of 3 nm to 10 mu m, Front Env Sci Eng, 10, 63-72, https://doi.org/10.1007/s11783-014-0754-x, 2016.

Wang, Z. B., Hu, M., Sun, J. Y., Wu, Z. J., Yue, D. L., Shen, X. J., Zhang, Y. M., Pei, X. Y., Cheng, Y. F., and Wiedensohler, A.: Characteristics of regional new particle formation in urban and regional background environments in the North China Plain, Atmos Chem Phys, 13, 12495-12506, 10.5194/acp-13-12495-2013, 2013.

Ma, L., Zhu, Y., Zheng, M., Sun, Y., Huang, L., Liu, X., Gao, Y., Shen, Y., Gao, H., and Yao, X.: Investigating three patterns of new particles growing to the size of cloud condensation nuclei in Beijing's urban atmosphere, Atmos Chem Phys, 21, 183-200, 10.5194/acp-21-183-2021, 2021.

Man, H. Y., Zhu, Y. J., Ji, F., Yao, X. H., Lau, N. T., Li, Y. J., Lee, B. P., and Chan, C. K.: Comparison of Daytime and Nighttime New Particle Growth at the HKUST Supersite in Hong Kong, Environ Sci Technol, 49, 7170-7178, 2015.

Shen, X., Sun, J., Kivekäs, N., Kristensson, A., Zhang, X., Zhang, Y., Zhang, L., Fan, R., Qi, X., Ma, Q., and Zhou, H.: Spatial distribution and occurrence probability of regional new particle formation events in eastern China, Atmos Chem Phys, 18, 587-599, 10.5194/acp-18-587-2018, 2018.

Lampilahti, J., Manninen, H. E., Nieminen, T., Mirme, S., Ehn, M., Pullinen, I., Leino, K., Schobesberger, S., Kangasluoma, J., Kontkanen, J., Järvinen, E., Väänänen, R., Yli-Juuti, T., Krejci, R., Lehtipalo, K., Levula, J., Mirme, A., Decesari, S., Tillmann, R., Worsnop, D. R.,

Rohrer, F., Kiendler-Scharr, A., Petäjä, T., Kerminen, V.-M., Mentel, T. F., and Kulmala, M.: Zeppelin-led study on the onset of new particle formation in the planetary boundary layer, Atmospheric Chemistry and Physics Discussions, 10.5194/acp-2021-282, 2021.

Paasonen, P., Peltola, M., Kontkanen, J., Junninen, H., Kerminen, V.-M., and Kulmala, M.: Comprehensive analysis of particle growth rates from nucleation mode to cloud condensation nuclei in boreal forest, Atmos Chem Phys, 18, 12085-12103, 10.5194/acp-18-12085-2018, 2018.

Zhang, J., Chen, Z., Lu, Y., Gui, H., Liu, J., Wang, J., Yu, T., and Cheng, Y.: Observations of New Particle Formation, Subsequent Growth and Shrinkage during Summertime in Beijing, Aerosol Air Qual Res, 16, 1591-1602, 10.4209/aaqr.2015.07.0480, 2016.

---

## Author Response (AR1)

Measurement Report: New particle formation characteristics at an urban and a mountain station in Northern China by Ying Zhou et al.

In this file, the referee comments are in black, our item-by-item replies are in blue, and the corresponding modifications in the manuscript are in red.

**Answers to reviewer #1**

NPF is a hot atmospheric topic and its air quality, climate and human health effect still remains not clear. As a measurement report, only 25 days are available, which cannot provide robust statistical results of NPF parameters (frequency, GR and FR). The author compared the NPF parameters with the previous studies in NCP region, however, the short period study can't explain the difference, but cause large uncertainties in statistical values. In this work, something new about the NPF events study should be pointed out as many similar studies have been conducted in the same region in China.

As per suggestion of the referee, we extended our data set at urban Beijing (UB) to include two full summers (from June 1 to August 31) in both 2018 and 2019. With the extended data, we updated our discussion on favorable and limiting factors for NPF event occurrence at the urban Beijing station. We also estimated the size of the area within which regional NPF events occurred during our observations. This estimation could possibly explain the phenomenon that most ending diameters of NPF events during our observation were limited to below 70 nm. Our results highlight the importance of anthropogenic emissions for NPF occurrence and subsequent growth in summer Beijing. Unfortunately, we were not able to include more data from the background mountain station (MT), since no long-term measurements were maintained. Yet, we are able to observe the contribution of the MT in forming particles and its difference from the UB station. Our study highlights the importance of establishing and maintaining long-term measurement not only in urban locations but also in rural ones. Major concerns:

1. Statistical significance: The statistical numbers of formation rate, growth rate, etc. was calculated based on only a few NPF cases (12, 13 NPF events at different locations). Is this statistical meaningful as the small quantity of cases? Also the comparison of the NPF frequency, GR, FR, and CS with the previous long-term study should be careful.

As mentioned above and as per suggestion of the referee, we extended our data set at urban Beijing station (UB) to include two full summers (from June 1 to August 31) in 2018 and 2019 to improve our statistical conclusions. With the extended data, we found that the shorter observation period between June 14 to July 14, 2019 is representative of the urban Beijing summer as no significant difference in NPF events' properties were observed when comparing the observation from June 14 to July 14, 2019 to the two full summer periods in 2018 and 2019. We then updated the comparison of particle growth rate, formation rate and CS based on the data measured in summer 2018 and 2019 at UB station.

The following discussion and figures (Figure R1-1, Figure R1-2 and Figure R1-3) will be added to the manuscript:

Figure R1-1a shows the difference in CS between NPF event and non-event days during our observation in summer 2018 and 2019 (two whole summers) at UB site and short-term parallel observations at both sites. The 'NPF1' and 'non-event1' referred to NPF and non-event days during the two whole summers, respectively, while 'NPF2' and 'non-event2' referred to NPF and non-event days during the short-term parallel observation period from June 14 to July 14, 2019 at both sites, respectively. The longer-term periods are used for confirming the representativeness of the short-term overlapping period for the whole summer. As shown in the figure, the median CS on NPF1 or NPF2 days is equivalent for UB station ( $CS_{NPF1} = 0.010s^{-1}$ ;  $CS_{NPF2} = 0.009s^{-1}$ ) and less than a factor of 1.2 different between non-event1 and non-event2 in UB station ( $CS_{nonevent1} = 0.023s^{-1}$ ;  $CS_{nonevent2} = 0.020s^{-1}$ ), which confirms the representativeness of our short-term measurement period to the overall urban Beijing summer.

Our results in Fig. R1-1a show that on NPF event days, the median CS was ~  $0.01 \text{ s}^{-1}$  during the first 2 hours of the NPF events, at both stations. On common NPF event days, the median

CS was 0.009 s-1 at UB station and ~ $0.01s^{-1}$  at MT station, respectively. In comparison, on non-event days, during roughly the same time period (9:00–11:00 LT), the CS was substantially higher, with median values of 0.02 s-1 and 0.014 s-1, at UB and MT stations, respectively. Figure R1-1b presents the median CS during the first 2 hours of NPF events on common NPF event days measured at both stations, and shows the high correlation between the two.

Figure R1-1c shows the NPF event frequency as a function of CS during our observation at UB site in summer 2018 and 2019 and how the NPF event frequency decreased with increasing CS. When CS was smaller than 0.01 s-1, all days were classified as NPF event days, and when CS was larger than 0.035 s-1, no day was classified as NPF event day. This shows the major role of background particles in controlling the occurrence or inhibition of NPF events as shown in several previous studies in China and internationally (Deng et al., 2020a; Cai et al., 2017; Kulmala et al., 2017). While we cannot present a similar figure from the MT station, the same conclusion applies where CS does play a role in inhibiting NPF observation owing to the difference in the CS values observed between NPF and nonevents at MT in Fig. R1-1a. Yet, since the overall preexisting particle concentration at the MT is rather on the low end, the role of CS might not be as vital at the MT station as for the UB station.

The particle formation rates ( $J_7$ ) at the two stations during the measurements are presented in Fig.R1-2a.  $J_7$  observed during the short-term parallel observation (NPF2) at UB site was in the range of 3.0-10.0 cm-3 s-1 with a median of 5.4 cm-3 s-1, comparable with those observed in summer 2018 and 2019 (NPF1 = 2-14.0 cm-3 s-1 with a median of 4.9 cm-3 s-1) and significantly higher than the values in the MT station (0.75-3.0 cm-3 s-1 with a median of 0.82 cm-3s-1) for common NPF events (Fig.R1-2b).

The particle growth rates in size range of 7-15 nm (GR7-15nm) at the UB station (4.8-12.9 nm/h with a median of 7.8 nm/h) during NPF2 was also comparable with the whole summers (NPF1) (4.8-12.9 nm/h with a median of 8.5 nm/h). While the difference in  $J_7$  was 7 times higher in UB than in MT, the observed GR were only slightly higher at UB than at the MT station (5.7-10.5 nm/h with a median of 6.5 nm/h) for common NPF events (Fig. R1-2c&d), implying that precursors needed for particle formation were much more abundant in the

polluted urban environment (Wang et al., 2013), while those needed for growth are rather comparable.

2. Instrument consistency: in the section 2.2, there is DMPS, SMPS, FMPS used in the PNSD measurement, the comparison of PNSD derived by different instrument should be given of the overlap size range. It is very important to make sure the data are comparable, as the PNSD data also determine the formation rate, growth rate and CS. The type and manufacturer of DMA of DMPS, as well as the SMPS should be also provide.

We thank the reviewer for the suggestions.

*The following discussion and figures (Figure R2-1) will be added to the manuscript in section 2.2:*

The DMPS consists of consists of one Hauke-type DMA (differential mobility analyzer, home-built by university of Helsinki) in different flow rates and one CPC (condensation particle counter, TSI Model 3772). Details of this instrument can be found in Salma et al., (2011) and Kangasluoma et al. (2020).

At MT station, a scanning mobility particle sizer (SMPS, consists of a TSI Differential Mobility Analyzer model 3081) and a fast mobility particle sizer (FMPS, TSI Model 3091) were used to measure particle number size distribution from June 14 to June 28 and from June 29 to July 14, respectively. The particle number size distribution measured by FMPS correlated well with SMPS after being calibrated (Lee et al., 2013).

To ensure high quality of particle number size distribution data at UB site, a particle number size distribution system (PSD) also sampled in parallel with DMPS from June 1 to August 31, 2019 (summer 2019). It measured particle number size distribution in the size range of 1 nm to 10  $\mu$ m. It included a nano-scanning mobility particle sizer (nano-SMPS, 3–55 nm, mobility diameter), a long SMPS (25–650 nm, mobility diameter) and an aerodynamic particle sizer (APS, 0.55–10  $\mu$ m, aerodynamic diameter). Details of this instrument can be seen at Liu et al. (2016) and Deng et al. (2020b).

As shown in Fig. R2-1, median particle number size distribution obtained from PSD and DMPS matched well with each other within a factor of 2 during our observation in summer

2018 and 2019 at UB site. We cannot compare particle number size distribution data obtained from DMPS, SMPS and FMPS as we did not sample with these three instruments in parallel at the same site. However, it is reasonable to assume that particle number size distribution obtained from FMPS were comparable with those from DMPS as on one hand the measurement techniques of particle number size distribution in the size range of these two instruments have been well developed and are applied in quite a lot observations in several environments (Wang et al., 2017;Kangasluoma et al., 2020). On the other hand, the FMPS was carefully calibrated and properly operated during the observation as discussed above. Similar conclusions apply for the SMPS as well, where we can rely on using the measurement from this instrument to discuss at least NPF event frequency at MT site during June 14 to June 28, 2019, during which parameters of only one NPF event are calculated.

3. The influence of air mass origin on the regional NPF occurrence was discussed. As the MT site locates nearby the mountain, how does the topography affect the air mass, local wind, as well as the inhomogeneity of regional NPF events should be also addressed.

We thank the referee for the suggestions. As per suggestions of the referee, the following discussions and figures (Figure R3-1 and Figure R3-2) are added to the manuscript results section:

**3.5 Effect of topography**

In Figure R3-1 we show average particle number size distribution and particle number concentration on NPF event and non-event days during our short-term parallel observation at both sites. On NPF event days, nucleation and Aitken mode particle number concentrations were much smaller at MT station than those at UB station due to smaller particle formation rates and less anthropogenic emissions. Interestingly, accumulation mode particle number concentrations were higher at MT station (701-2900 cm-3, with a median of 1500 cm-3) than that at UB station (350-1416 cm-3 with a median of 700 cm-3) (Fig.R3-1b). Due to the close proximity of the two measurement sites, the air mass arrival directions and source regions were (mostly) similar at both sites throughout the short-term parallel measurement period (Table 1) hence the regional and transported cannot explain the higher accumulation mode

particle number concentration at MT site. As there were few primary emissions at MT site, the accumulation mode particles could be attributed to secondary particles (Kulmala et al., 2021), indicating particles at MT station were more aged than those at UB station (Fig.R3-1a). The possible reason is that mountains block pollution diffusion, which in the end resulted in comparable CS at MT station as UB station.

Figure R3-2 shows an example of the wind distribution before and during NPF event on June 30, 2019 at 850 hPa (close to the altitude of MT station) and 10 m above ground level. As shown in Fig. R3-2, the reanalyzed wind directions at 850 hPa were similar as those at 10 m above the ground level at MT station. Actually, the wind conditions on other NPF event days at MT station during our observation had similar characteristics that the wind directions were similar between 850 hPa and 10 m above ground level indicating air masses well mixed during NPF events. Earlier observations also found NPF event happened uniformly within the mixing layer at their observation stations and particle number size distribution remains roughly constant within the mixing layer (Shen et al., 2018;Lampilahti et al., 2021).

**Minor comments:**

1. L18-19, I don't think this conclusion is appropriate in the abstract, as the sentence imply this is the first work about urban and regional measurement. Actually, Wang et al., (2013) has reported the regional NPF events in urban Beijing and a regional background site based on one-year dataset before.

We thank the referee for the comments. We agree that our work about urban and regional measurement is not the first but the observations on NPF events on the mountain is still rare in China emphasizing the need of establishing and maintaining such long term observations there.

**As per suggestions of the referee, we corrected the sentence in line 18-19 as below:**

Most observations on NPF events in Beijing and its vicinity were conducted in populated areas, whereas observations on NPF events in mountain sites with few anthropogenic emissions are still rare in Beijing (Wang et al., 2013). The spatial variation of NPF event intensity has not been investigated in detail by incorporating both urban area and mountain measurements.

**2. L47, the reference of the same author should be cited as Guo et al., 2014; 2020.**

We corrected the citation as Guo et al., 2014; 2020 in our manuscript.

**3. L217, what is "good data"? The clear explanation should be given.**

The "good data" meant that particle number size distribution were valid that visual inspection of the data and the number concentrations as well as instrument status do not indicate problems in the measurements at both sites. As per suggestions to the reviewer, we corrected sentence in line 217 as below:

Only days when particle number size distribution data were valid that visual inspection of the data and the number concentrations as well as instrument status do not indicate problems in the measurements for both stations were taken into consideration in our analysis.

4. L222, higher NPF frequency in this study, as compared with Wang et al., 2013; Deng et al., 2020a, was explained by "25 days were validated data", it is not convincing. The short period study caused large uncertainty in the comparison, including the formation rate, growth rate, frequency and CS. However, it can't explain for the higher or lower value. Other favorable parameters for NPF, e.g. meteorology, precursors, CS, should also be taken into consideration.

**We thank the referee for the comments. As per suggestions to the reviewer, the following discussions and figures (Fig.R4-1, Fig.R4-2 and Fig.R4-3) will be added in our manuscript:**

In Fig.R4-1, we show frequencies of air masses arriving at UB station from different directions during our observation in summer 2018 and 2019. The most frequent air masses arriving at UB station belonged to the South group. During our observation in the two summers, out of 155 days were 52 days belonging to the South group and 39, 32, 9 and 23 days in air masses belong to North, East, West and Local groups, respectively. NPF event frequency with respect to air masses is also shown in Fig. R4-1. It is noticeable that air mass origin influenced the occurrence of NPF events at UB site as the majority of NPF events occurred when the air masses were coming from the north. During our observation in summer 2018 and 2019, 34 (out of 55) NPF events occurred in air masses from the North group and 9, 2, 2 and 6 NPF events in the South, East, West and Local groups, respectively (Fig.R4-1a). One prominent feature of these air masses is their difference in CS. As shown in Fig. R4-1b,

the CS of the air masses classified as the North group (with median values of 0.01 s-1 at UB station) is substantially lower than that in other air mass classes (CS = 0.03, 0.025, 0.017, 0.03 s-1, for south, east, west and local, respectively), which might explain the high NPF event frequency associated with this air mass class. During the observation from June 14 to July 14 in summer 2019, the most frequent air masses arriving at both sites belonged to the North group as shown in Table 1. Out of 25 days, there were 8 and 9 days belonging to the North group, at UB and MT sites, respectively. The highest frequency of NPF events also occurred when the air masses were coming from the north. The high NPF events frequency during our observation form June 14 to July 14 could also be attributed to the frequent air masses arriving at both sites from north to Beijing.

In Figure R4-2, we show diurnal variation of meteorological variables during our observation in summer 2018 and 2019 at UB site and observations from June 14 to July 14 in 2019 at UB and MT sites. It is noticeable that the short-term observation compared well with the long-term observation and therefore is representative of summer at UB site as shown in Fig.R4-2.

In Fig. R4-3a, we show the concentration of sulfuric acid as a function of CS during summer 2018 and 2019 at UB site. As shown in Fig. R4-3b, the median sulfuric acid (H2SO4) concentrations at UB station were  $8.1 \times 10^6$  cm-3 and  $4.5 \times 10^6$  cm-3 on NPF event days and non-event days, respectively, during observation from June 14 to July 14 in 2019 and  $7.9 \times 10^6$  cm-3 and  $3.4 \times 10^6$  cm-3 on NPF event days and non-event days, respectively, during the observation in summer 2018 and 2019. This suggests that H2SO4 was important for NPF events at the UB station (Deng et al., 2020b; Dada et al., 2020b). On the other hand, as shown in Fig.R4-3a, the H2SO4 concentration during 9:00- 11:00 (local time) on non-event days could be comparable with that on NPF event days, especially when CS was high. Altogether, our observation shows that the occurrence of NPF events was controlled by both H2SO4 and CS at the UB station (Cai et al., 2020).

5. Section 3.1.4, do you mean the higher ending diameter at UB site, supported by the higher condensing vapors? But as you have mentioned, the GR at both sites were comparable, does that mean the condensing level are also comparable? The conclusions from GR and ending

**diameter were not consistent, it should be discussed further.**

We thank the referee for the comments. We found that the ending diameters were slightly higher at UB site than at MT site, but the difference is not significant (49 nm vs 45 nm), in addition, the GR on common NPF event days were a little bit higher at UB site than MT site. Earlier research pointed out that in order to observe the growth until 100 nm at the measurement station under typical conditions, simultaneous NPF should happen in a very large area (e.g. with wind speed 5m/s and growth rate of 3nm/h from the station to roughly 600 km upwind from the station) (Paasonen et al., 2018). To discuss the ending diameters, we evaluated the size of regions that NPF events could be occurring during our observation. Also, as per suggestions of the reviewer, we updated our comparison of GR on common NPF events as well as the discussion on ending diameters as follows:

The particle growth rates in size range of 7-15 nm (GR7-15nm) at the UB station (4.8-12.9 nm/h with a median of 7.8 nm/h) during observation from June 14 to July 14, 2019 was also comparable with that during the observation in summer 2018 and 2019, a little bit higher than that in the MT station (5.7-10.5 nm/h with a median of 6.5 nm/h) for common NPF events (Fig.R1-2c&d), implying that precursors were only slightly more abundant in the polluted urban environment (Wang et al., 2013).

Addition to the methods' section:

**2.4 Estimating the spatial extent of NPF**

The observation of regional new particle formation events, where the growth of newly formed particles can be followed for several hours, is a result of NPF taking place over a large spatial area. This is because as time progresses, the particles observed at a measurement site must have originated from further and further away due to non-zero wind conditions. Following the progression of the observed NPF event and using air mass back trajectories, we can estimate where the particles observed at different stages of the NPF event were initially formed by calculating the air mass locations at the onset time of the NPF event (assuming that NPF occurs simultaneously over the larger area). Typically, the mode related to the NPF event disappears from the observations after some time. This is an indication of the currently observed air mass arriving from an area where NPF was no longer taking place due to unfavorable local conditions. If the shift in the air mass origin towards unfavorable conditions occurs gradually over time, the mode related to the NPF event can enter a stage of growth stagnation (or even decrease in size) before disappearing completely (Kivekäs et al., 2016). This is because the increasing transport time between NPF onset and observation of the particles at the measurement site provides less and less additional 'material' for aerosol growth towards the more unfavorable conditions. Calculating the locations where NPF is assumed to have taken place for longer data sets including several regional NPF events can give an estimation of the typical spatial extent of NPF around the measurement location. It should be noted that even in relatively clear cases, the subjective determination of NPF event onset and end times can easily lead to uncertainties of few tens of kilometers in the estimations. In locations with strong primary pollution sources, such as urban Beijing, objective determination of said times becomes even more difficult. More details and discussion related to the method and its uncertainties can be found in Kristensson et al. (2014).

Results of the NPF spatial analysis are shown in Figure R5-3 and discussed in section 3.4 below.

Addition to Results section:

**3.4 Ending diameters of newly-formed grown particles**

[revised manuscript text omitted]

6. Figure 1: why the station of S60 is shown in the figure?

We thank the referee for the comments. The S60 referred to the location where particles formed during the non-local NPF event observed at MT station on June 15, 2019.

We added the following sentence in our manuscript:

**Figure 1:** Map showing locations of urban station (UB), Longquan station (LQ), mountain station (MT) and another site 60 km south from MT station (S60). The S60 referred to the location where particles formed during the non-local NPF event observed at MT station on June 15, 2019. Image is produced using © Google Maps.

7. The key word: haze, there is no much discussion about how NPF event contributing to regional haze formation in the study. So I think this key word is not appropriate.

We thank the referee for the comments. As per suggestion to the reviewer, we removed the key word "haze".

**Figures**

**Figure R1-1:** (a) Median and percentiles of condensation sink (CS, s-1) during our observations at both stations. The 'NPF1' and 'non-event1' referred to NPF and non-event days during summer 2018 and 2019, while 'NPF2' and 'non-event2' referred to NPF and non-event days during the short-term parallel observation from June 14 to July 14, 2019 at both sites. The red line represents the median of the data and the lower and upper edges of the box represent  $25^{th}$  and  $75^{th}$  percentiles of the data, respectively. The length of the whiskers represents  $1.5 \times$  interquartile range which includes 99.3% of the data. The time resolution of CS was 8 min. (b) Median CS during the first 2 hours of NPF events on common NPF event days measured at both stations (MT vs. UB). (c) Numbers of NPF event, non-event and undefined days as well as NPF event frequency as a function of CS during our observation in summer 2018 and 2019 at UB station.

---

## Referee Report (RR1)

The authors extend the dataset and the paper has been improved significantly. However, there are some issues that the author should give more explainations. There are also some grammatical and spelling errors in manuscript. The authors need to check through the manuscript carefully.

Major concern:
1. The accumulaiton mode particles number was reported to be ~700 cm$^{-3}$ on NPF days on urban site, which is almost 50% lower than the mountain site. The author should check the data, as it was also reported the CS on NPF days at both sites was ~0.01 s$^{-1}$. As the CS values are quite similar and dominated by the accumulaiton mode, why the difference in acumulaiton mode concentration between the two sites is so large? Also, it should be clear that the size range of each mode in the text.
2. In the summer campaign, which is rainy season in Beijing, the precipitation should be addressed as it is an important scavenging process of particles. Also, for Mounatin site, the fog/cloud process is another particle scavanging process that can influence the CS.

Sepcific comments:
1. Line 27, it's not necessary to give the reference in the abstract.
2. Line 38-40, The author addressed the CS during the first two hours of NPF, indicating the concentration level of the pre-existing particles. However, why the formation rates during the first two hours of NPF are compared?
3. Line 110, please shorten "particle formation rate, particle growth rate" to "formation rate and growth rate". Check these words through the manuscript.
4. Line 125, the site S60 and LQ should be illustrated in the site discribiton, not in the figure caption. In the discussion, S60 station is not mentioned.
5. Line 138, it should be Salma et al., (2011)
6. Line 156-157, PNSD meausred by PSD and DMPS matched with a factor of 2…it is not clear which PNSD is higher? The ratio of 2 is derived by the total number concentration or what else? In Fig.3, it seems the PSD is higer below 20 nm and 300-600 nm, but lower above 600 nm. Which data is referred as the true value?
7. Line346, As shown in Figure ?? which figure? Line 351, in Figure 8a. The initial letter of Table and Figure should be capitalized. Please check all through the manuscript. Figure and Fig are both used, it should be consistent.
8. Section 3.4 I suggest to revise the end Dp as $D_{p,end}$, also be consistent with figure caption.
9. I suggeted each subplot in Fig.14 and other figure pannels should be marked as a, b, c, d,…, in order to be referred easily. Please also check the figure captions as some are not complete, e.g., Fig. 18, the marker of mode diameters is not given.
10. The value of CS is suggested to be added in Table 1.
11. Spelling check (including the below, but not limited to):
    Line 112, "favorable conditions";
    Line 114, conditions those… could help to minimize…

Line 295, NPF characteristics;

Line 239, said times??

Line 388, is considered to be one of the most…

Line 511, please check Fig. R12c

Line 568, This is a common…

---

## Author Response (AR2)

*Measurement Report: New particle formation characteristics at an urban and a mountain station in Northern China by Ying Zhou et al.*

In this file, the referee comments are in black, our item-by-item replies are in blue, and the corresponding modifications in the manuscript are in red.

**Answers to reviewer # 1**

The authors extend the dataset and the paper has been improved significantly. However, there are some issues that the author should give more explanations. There are also some grammatical and spelling errors in manuscript. The authors need to check through the manuscript carefully.

We would like to thank the referee for their suggestions and careful editorial comments. We present our answers to the referee comments point-by-point below and paid careful attention to the grammatical and spelling errors in manuscript.

Major concerns:

1. The accumulation mode particles number was reported to be ~700 cm$^{-3}$ on NPF days on urban site, which is almost 50% lower than the mountain site. The author should check the data, as it was also reported the CS on NPF days at both sites was ~0.01 s$^{-1}$. As the CS values are quite similar and dominated by the accumulation mode, why the difference in accumulation mode concentration between the two sites is so large? Also, it should be clear that the size range of each mode in the text.

We revisited our data and calculation of CS as well as particle number concentration of every mode at both sites.

As shown in Figure R2-1 (c&e) on NPF event days, particles smaller than 100 nm contribute to a CS of $3.7\times10^{-3}$ s$^{-1}$, contributing 37% to the total CS. While at MT site, particles smaller than 100 nm only contribute to a CS of $1.2\times10^{-3}$ s$^{-1}$, contributing less than 12% to the total CS (Figure R2-1 (d&f)). Although 100-840 nm particle number concentration at UB site was

much less than that at MT site, 1-100 nm (especially 25-100 nm) particles compensated total CS by higher number concentration on NPF event days.

The nucleation, Aitken and accumulation mode particles we mentioned in our manuscript were particles in the size ranges of 6-25 nm, 25-100 nm and 100-840 nm, respectively.

As per suggestion by the reviewer, we updated Figure 6 in the manuscript as Figure R2-1 and add discussion on CS in line 264 as below:

As shown in Figure 6 (c&e) on NPF event days, particles smaller than 100 nm contribute to a CS of $3.7 \times 10^{-3}$ s$^{-1}$, contributing 37% to the total CS. While at MT site, particles smaller than 100 nm only contribute to a CS of $1.2 \times 10^{-3}$ s$^{-1}$, contributing less than 12% to the total CS (Figure 6 (d&f)). Although 100-840 nm particle number concentration at UB site was much less than that at MT site, the 1-100 nm (especially 25-100 nm) particles largely participated by higher number concentration on NPF event days to result in a comparable CS between both sites.

[Figure]

Figure R2-1: Median CS size distribution (a&b), accumulated CS contributed by particles

from 6 nm and the ratio between accumulated CS and total CS (c&d); Contribution of size-segregated particles to total CS (e&f) at each site on NPF and non-event days during 9:00-15:00 (local time, LT). Figures on the left and right panels represented data observed at UB and MT site, respectively. The time resolutions for CS and particle number concentration data were 8 min at UB station and 4 min at MT station, respectively.

As per suggestion by the reviewer, we added size range of each mode in line 534-536 as below:

On NPF event days, nucleation (6-25 nm) and Aitken (25-100 nm) mode particle number concentrations were much smaller at MT station than those at UB station due to smaller formation rates and less anthropogenic emissions. Interestingly, accumulation (100-840 nm) mode particle number concentrations….

2. In the summer campaign, which is rainy season in Beijing, the precipitation should be addressed as it is an important scavenging process of particles. Also, for Mountain site, the fog/cloud process is another particle scavenging process that can influence the CS.

We thank the reviewer for the comments. In summer 2018 and 2019, the DMPS data were discarded from analysis on rainy days as rain affected the quality of the DMPS data. We calculated CS at both sites assuming RH equals to 0%. When we calculated the RH affected CS at MT site, e.g. 30%-70% during 9:00-15:00, as shown in Figure 10, the CS would be 1.12-1.33 times of those with a RH of 0%, as shown in Figure R2-2.

As per suggestion to the reviewer, we added the following discussion in line 264:

The data on rainy days were discarded from analysis at both sites, hence the precipitation was considered to have minor effects on our CS calculation. We calculated CS at both sites assuming RH as 0%. It should be noted that the CS may have been underestimated by a factor of 1.12-1.33 at MT site when we include RH in the CS calculation, e.g. 30%-70% during 9:00-15:00, as shown in Figure 10.

[Figure]

Figure R2-2: CS calculated with RH at MT site (30%, 50% and 70%) as a function of CS calculated with RH as 0%. The slopes of the fitted lines represent ratios between CS with RH at MT site and RH as 0%.

Specific comments:

1. Line 27, it's not necessary to give the reference in the abstract.

We removed the reference from the abstract.

2. Line 38-40, the author addressed the CS during the first two hours of NPF, indicating the concentration level of the pre-existing particles. However, why the formation rates during the first two hours of NPF are compared?

We thank the reviewer for the suggestions. Determination of nucleation start and stop times were difficult at UB site due to the contribution of traffic emissions. Hence, we choose a time window of the first 2 hours of NPF event for formation rates calculation at both sites for objectivity. During the time window, we always observed 7-10 nm particle number concentration burst significantly from the background level at both sites.

According to the comments of the reviewer, the following sentence will be added in our manuscript of line 286:

Determination of nucleation start and stop times was affected by traffic emissions at UB station. Hence, we choose a time window of the first 2 hours of NPF event for formation rates calculation at both sites. During the time window, we always observed 7-10 nm particle

number concentration burst significantly from the background level at both sites.

3. Line 110, please shorten "particle formation rate, particle growth rate" to "formation rate and growth rate". Check these words through the manuscript.

Corrected.

4. Line 125, the site S60 and LQ should be illustrated in the site description, not in the figure caption. In the discussion, S60 station is not mentioned.

We thank the reviewer for the suggestion. As suggested by both reviewers, the S60 site has been removed from the discussion as well as the map in our manuscript. The introduction of LQ station has been added to the method's section as below:

Longquan station: The Longquan national monitoring station sits in Longquan town, Mengtougou District, Beijing. It is 20 km west to UB site and 60 km east to MT site and considered a suburban station. The location is referred to as '**LQ**' from here after and is shown on the map in Figure 1.

5. Line 138, it should be Salma et al., (2011)

Corrected.

6. Line 156-157, PNSD measured by PSD and DMPS matched with a factor of 2…it is not clear which PNSD is higher? The ratio of 2 is derived by the total number concentration or what else? In Fig.3, it seems the PSD is higher below 20 nm and 300-600 nm, but lower above 600 nm. Which data is referred as the true value?

We thank the reviewer for the comments. We chose PSD data as reference, e.g. PSD data is referred as true value. The ratio of 2 is derived by particle number size distribution. As per suggestion by the reviewer, we corrected the sentences in Line 156-157 as following:

The PSD was used as reference. As shown in Figure 3, particle number size distribution measured by DMPS matched well with PSD in data trend. Varying with particle diameter, particle number size distribution data measured by DMPS can be higher or lower than PSD within a factor of 2.

7. Line346, As shown in Figure ?? which figure? Line 351, in Figure 8a. The initial letter of Table and Figure should be capitalized. Please check all through the manuscript. Figure and

Fig are both used, it should be consistent.

We checked all through the manuscript for consistency and corrected Line 346:

As shown in Figure 8a,…

As per suggestion by the reviewer, we used Figure in our manuscript and changed all "Fig" into "Figure".

8. Section 3.4 I suggest to revise the end Dp as $D$p,end, also be consistent with figure caption.

We thank the reviewer for the suggestions.

To avoid ambiguity, we changed the figure caption in the manuscript as following:

[Figure]

Figure 12: (a) Median and percentiles of end diameters (End Dp, nm) of NPF events measured at both sites. The red line represents the median of the data and the lower and upper edges of the box represent 25th and 75th percentiles of the data, respectively. The length of the whiskers represents 1.5× interquartile range which includes 99.3% of the data. The 'NPF1' and 'non-event1' referred to NPF event and non-event days in summer 2018 and 2019 and the 'NPF2' and 'non-event2' referred to NPF event and non-event days during the observation from June 14 to July 14, 2019. (b) Frequencies of end diameters in the size range of smaller than 25 nm, 25-70 nm, 70-100 nm and above 100 nm during our observation at UB

station in summer 2018 and 2019. (c) Comparison between end diameters of coincident NPF
events at both stations.

9. I suggested each subplot in Fig.14 and other figure panels should be marked as a,

b, c, d,…, in order to be referred easily. Please also check the figure captions as some are not

complete, e.g., Fig. 18, the marker of mode diameters is not given.

We thank the reviewer for the comments. As per suggestion by the reviewer, we corrected
Figure 14 as bellow.

[revised manuscript text omitted]

In this file, the referee comments are in black, our item-by-item replies are in blue, and the corresponding modifications in the manuscript are in red.

**Answers to reviewer # 2**

This work reported simultaneous measurements of new particle formation events during an intensive campaign at an urban and a mountain station in China. It is a complex and extended study that fits well with to the scope of ACP and it is of interest for the international research community. However, there are some issues to be improved or corrected before it is published in ACP.

We would like to thank the referee for the suggestions and careful editorial comments.

We present our answers to the referee comments point-by-point below.

Major comments

1. The authors include a long paragraph in the introduction to summarize some results on the regional extension of NPF. This manuscript deals with the extension of NPF events but also with two different altitude sites (one mountain site). In my opinion, it is necessary to include a paragraph about the state-of-the-art of NPF events at mountain sites and the vertical distribution of NPF. Sellegri et al (2019) reviewed NPF events at mountain sites and it is not cited along the manuscript. This same manuscript also discusses the topography or preferred altitude of NPF events. In addition, there is some studies that attempt to look on the vertical distribution of NPF (e.g., Komppula et al., 2003; Boulon et al., 2011). Finally, there is also a recent and similar study that looks on the differences of NPF at two sites, also urban and mountain sites, (Casquero-Vera et al., 2020) and the differences or similarities of the results should be discussed with this similar study where the GR, J or CS are also discussed at two different altitude sites.

We thank the reviewer for the suggestions. We added a paragraph between line 104 and line

In addition to horizontal extension of NPF events, the vertical extension of NPF events also attracts attention of researches. It have been confirmed that NPF events can be triggered within the whole low tropospheric column at the same time and even above the planetary boundary layer upper limit (Boulon et al., 2011). Sellegri et al. (2019) reviewed NPF events observed at 6 different altitude stations. They found NPF events were most favored at the altitude close to the interface of the free troposphere (FT) with the planetary boundary layer (PBL) and at the vicinity with clouds. In addition, at high altitude sites, CS may not be the liming factor for NPF occurrence as higher CS associated with more precursors for nucleation and initial growth. Based on observations at two different altitudes (e.g. 340 m and 560 m above sea level) in northern Finland, Komppula et al. (2003) found NPF events had similar formation and growth rates between these two heights, while due to vertical movement of air masses, difference of NPF event start time between these two sites was limited within 30 min. Similar results were also observed at two sites in France that formation and growth rates were similar between two altitudes (e.g., 660 m and 1465 m above sea level) while the contribution of ion-induced nucleation was higher at high altitude (Boulon et al., 2011). Finally, during a recent observation in Spain, growth rates were higher at the mountain site (2500 m a.s.l.) than urban site (680 m a.s.l.), while difference between formation rates varied with altitude (Casquero-Vera et al., 2020).

2. In P19 the authors stated that urban emissions affect the formation rates, but the NPF are of regional extension? Could local events happen without that regional phenomena? Could the emissions of that huge city be the unique responsible of the regional NPF? In this same section, the authors suggest that "precursors needed for particle formation were much more abundant in the polluted urban environment (Wang et al., 2013), while those needed for growth are rather comparable". The analysis of J is for 7nm size, that means these particles are not newly formed, these particles come from growth or could be emitted directly by i.e. traffic? At MT, 7 nm particles means that these particles could not be formed there or the vicinity? Please clarify these ideas.

We thank the reviewer for the comments. As we discussed in section 3.4, the upwind extension of regional NPF events was limited to the areas with some anthropogenic emissions. There should not be any discrete boundary between the regions that NPF event is or is not occurring, but with decreasing anthropogenic emissions, the strength (formation rates and growth rates) are expected to decrease. Particle formation rates were usually positively correlated with $H_2SO_4$ concentration, the urban emissions can provide abundant $SO_2$, hence abundant $H_2SO_4$, resulting in high formation rate (Kerminen et al., 2018).

Local events can happen without regional phenomena. We sometimes observed nucleation mode particle number concentration burst without mode diameter increasing. It could be related to non-regional NPF events (Dai et al., 2017). We did not observe such cases at the MT site. Actually, the abundant anthropogenic emissions in the megacity could provide enough precursors for non-regional NPF events. However, traffic emissions can also provide abundant primary nucleation mode particles making it difficult to distinguish whether the new mode was from NPF event or traffic. So we classified such events as "undefined" also.

During our observation, there was no air mass convection between two sites. And we did not conduct chemistry measurement at MT site, as a result, the contribution of urban emissions on particle growth is unknown there. So we are not sure whether urban emissions can be the unique responsible of regional NPF events. To figure this question out, we still need long time observation on gas and particle phase chemistry as well as particle number size distribution down to sub-3 nm downwind urban Beijing.

Due to the non-zero wind conditions, the 7 nm particles we observed at both sites should have originated upwind the sites. In addition, Boulon et al. (2011) observed that new particles could be formed at low altitude and transported to the higher altitude sites. However, to confirm whether the phenomenon can happen at MT site, we still need observations on vertical wind conditions or vertical evolution of potential temperature. At UB site, the traffic emissions can also provide 7 nm particles, but compared with NPF events, the contribution of traffic emissions is considered minor (Zhou et al., 2020).

As per suggestion by the reviewer, we updated our conclusion in line 640 as below:

The upwind extension of regional NPF events was limited to the areas with some anthropogenic emissions. There should not be any discrete boundary between the regions that NPF event is or is not occurring, but with decreasing anthropogenic emissions, the strength (formation rates and growth rates) should decrease.

As per suggestion by the reviewer, we added such sentence below in line 251 as below:

At UB site, we also observed some cases in which nucleation mode particle number concentration burst with mode diameter increase. It could be related to non-regional NPF events (Dai et al., 2017). We did not observe such cases at the MT site. Actually, the abundant anthropogenic emissions in the megacity could provide enough precursors for non-regional NPF events. However, traffic emissions can also provide abundant primary nucleation mode particles, making it difficult to distinguish whether the new mode was from NPF event or traffic. So we classified such events as "undefined" also.

As per suggestion by the reviewer, we updated our conclusion in line 644 as below:

For more robust knowledge on NPF events in north China plain and to figure out the effect of urban emissions on regional NPF events, we still need long-term observations including particle number size distribution down to sub-3 nm, gas and particle phase chemistry downwind and upwind urban Beijing.

As per suggestion by the reviewer, we updated our conclusion in line 644 as below:

Also, the $J_7$ at UB station could be affected by traffic emissions due to the proximity of the location to the highway, while compared with NPF events, the effect of traffic emissions is shown to be minor (Kontkanen et al., 2020; Zhou at al., 2020). In addition, Boulon et al. (2011) observed that new particles could be formed at low altitude and transported to the higher altitude sites, however, to confirm whether the phenomenon can happen at MT site, we still need observation on vertical wind conditions or vertical evolution of potential temperature.

3. Case studies are "special cases" but, in my opinion, they are not analyzed in depth. For example, shrinkage cases are of interest since there is not clear the origin of this phenomenon (e.g., Salma et al., 2016; Alonso-Blanco et al., 2017). In this sense, Section 3.6.3 makes an

attempt to discuss the case of stagnant and shrinkage but unfortunately, there is not a real study or discussion of this special case. Please go further on this or remove these sections.

As per suggestion by the reviewer. With the data we have now, we are not able to provide much clarification on mode diameter shrinkage issue. So we removed the case studies part, e.g. section 3.6.

4. Please review the whole text, there is long sentences without any comma and not well connected along the manuscript.

As suggested by both reviewers, we have corrected all the grammatical and spelling errors in manuscript as much as we can find.

Minor comments

L26 – Change "few" to "low"

Corrected.

L41 – "at urban site" is repeated

Corrected.

L128 – And the altitude of the urban site?

We thank the reviewer for the suggestion. The altitude of the west campus of BUCT is around 50 m above sea level and the urban site is located on the fifth floor of a university building inside the west campus of BUCT, around 12 m above ground level.

According to the comments of the reviewer, the following sentence will be added in our manuscript of line 122:

The altitude of the west campus of BUCT is around 20 m above ground level and the urban site is around 12 m above ground level.

L138 – Cite format

Corrected.

L148-149 – Why do you mention Fig 2 here? Strictly, both instruments cannot correlate if they don't measure at same time.

We thank the reviewer for the suggestion. We mention Figure 2 here to show a general data quality of particle number size distribution during our short time parallel observation. The

FMPS matched well with SMPS in laboratory comparison after being well calibrated.

According to the comments of the reviewer, we removed the sentence in line 147 and add the following sentence below in line 134:

As shown in Figure 2, the data qualities of particle number size distribution at both sites during the short-term parallel observations was good in general.

According to the comments of the reviewer, we modified the sentence in line 148 in our manuscript as below:

The particle number size distribution measured by FMPS matched well with SMPS during the comparison in laboratory after being calibrated.

L160 – It "is" reasonable

Corrected.

L164 – What calibrations were done?

The particle number size distribution from FMPS were calibrated according to the method introduced by Zimmerman et al. (2015).

According to the comments of the reviewer, we modified the sentence in line 164 in our manuscript as below:

On the other hand, the particle number size distribution from FMPS was carefully calibrated and the FMPS was properly operated during the observation as we discussed above.

L181 – Space after dot

Corrected.

L221 – must have been?

We thank the reviewer. We corrected the sentence as following:

This is because as time progresses, the particles observed at a measurement site should have originated from further and further away due to non-zero wind conditions.

L278 – Is it correct that the "dk" is included and the "du" is not? "[dk, du)"

We thank the reviewer for the comments. $N_{[dk,du)}$ is defined as the total number concentration of particles in the size range from dk to du (particles with diameters of du are not accounted for) (Cai and Jiang, 2017).

According to the comment of the reviewer, we corrected L278 in our manuscript as below:

$N_{[dk,du)}$ is defined as the total number concentration of particles in the size range from dk to du (particles with diameters of du are not accounted for).

L288 – NPF event "frequencies" is not an adequate title for this section. Maybe something on the "occurrence" but not the frequency. I suggest "Origin of NPF events at both sites"?

We thank the reviewer for the good suggestion. According to the suggestion by the reviewer, we changed the title of this section as "Origin of NPF events at both sites".

L291-293 – Rephrase

We thank the reviewer for the suggestions. We rephrased the sentence in our manuscript as following:

The NPF event frequency was consistent with an earlier observation in summer in urban Beijing from 2004 to 2008, while smaller than other seasons especially winter during that observation and another one-year observation in UB station.

L300-303 – Rephrase, use comma.

We thank the reviewer for the suggestions. We rephrased the sentence in our manuscript as following:

Data were considered as valid when visual inspection of the particle number size distribution data and the instrument status did not indicate problems in the measurements. Only days with valid data at both stations were taken into consideration in our analysis.

L304 – "Common" is referred to "coincident" events? If it is, change the terminology.

We thank the reviewer for the suggestions. We changed "common events" into "coincident events" in our manuscript.

L615 – Mechanisms are not really investigated

We thank the reviewer for the comments.

We deleted "and mechanism" in line 615 in our manuscript.

References

[revised manuscript text omitted]